# ChAMBRe: a new atmospheric simulation Chamber for Aerosol Modelling and Bio-aerosol Research

Dario Massabò[1], Silvia Giulia Danelli[2], Paolo Brotto[3], Antonio Comite[4], Camilla Costa[4], Andrea Di Cesare[5], Jean François Doussin[6,] Federico Ferraro[1], Paola Formenti[6], Elena Gatta[7], Laura Negretti[4], Maddalena Oliva[4], Franco Parodi[2], Luigi Vezzulli[5], Paolo Prati[1]

[1]Dipartimento di Fisica - Università di Genova and INFN - Sezione di Genova, via Dodecaneso 33, 16146, Genova (IT)
[2]INFN - Sezione di Genova, via Dodecaneso 33, 16146, Genova (IT)
[3]PM_TEN srl, Piazza della Vittoria 7/14, 16121, Genova (IT)
[4]Dipartimento di Chimica e Chimica Industriale, Università di Genova, via Dodecaneso 31, 16146, Genova (IT)
[5]Dipartimento di Scienze della Terra dell'Ambiente e della Vita, Università di Genova, Corso Europa 26, 16132 Genova (IT)
[6]LISA, UMR CNRS 7583, Université Paris Est Créteil et Université Paris Diderot, Institut Pierre-Simon Laplace, Avenue du Général de Gaulle, 94000 Créteil (FR)
[7]Dipartimento di Fisica, Università di Genova, via Dodecaneso 33, 16146, Genova (IT)

*Correspondence to:* Dario Massabò (massabo@ge.infn.it)

**Keywords:** *atmospheric simulation chambers, bio-aerosol, Bacillus subtilis, Escherichia coli*

**Abstract.** Atmospheric simulation chambers are exploratory platforms used to study various atmospheric processes at realistic but controlled conditions. We describe here a new facility specifically designed for the research on atmospheric bio-aerosol as well as the protocols to produce, inject, expose and collect bio-aerosols. ChAMBRe (Chamber for Aerosol Modelling and Bio-aerosol Research) is installed at the Physics Department of the University of Genova, Italy, and it is a node of the EUROCHAMP-2020 consortium. The chamber is made of stainless steel with a total volume of about 2.2 m³. The lifetime of aerosol particle with dimension from a few hundreds of nanometres to a few microns varies from about 10 to 2 hours. Characteristic parts of the facility are the equipment and the procedures to grow, inject and extract bacterial strains in the chamber volume while preserving their viability. Bacteria are part of the atmospheric ecosystem and have impact on several levels as: health related issues, cloud formation, and geochemistry. ChAMBRe will host experiments to study the bacterial viability versus the air quality level, i.e. the atmospheric concentration of gaseous and aerosol pollutants. In this article, we report the results of the characterization tests as well as of the first experiments performed on two bacterial strains belonging to the Gram positive and Gram negative groups. A reproducibility at the 10% level has been obtained in repeated injections and collection runs with a clean atmosphere, assessing this way the chamber sensitivity for systematic studies on bacterial viability vs. environmental conditions.

# 1. Introduction

## 1.1 The problem of bio-aerosol and bacterial strains

The biological component of atmospheric aerosol (bio-aerosol) is a relevant subject of both atmospheric science and biology. From the pioneer investigations at the end of the nineteenth century (Pasteur, 1862), the study of primary biological aerosol particles (PBAP) has definitively become a multidisciplinary field of research, which requires expertise in physics, chemistry, biology and medical sciences (Desprès et al., 2012). Among PBAP, bacteria have a crucial role (Bowers et al., 2010). They show atmospheric concentrations from $10^4$ to $10^6$ cells $m^{-3}$ (Ligthart, 1997, 2000) with a wide range of diversity (Amato et al., 2007; Burrows et al., 2009; Gandolfi et al., 2013; Maki et al., 2013). Bacterial viability, including the capability of pathogens to survive in aerosol and maintain their pathogenic potential, depends on the interaction between bacteria and the other organic and inorganic constituents in the atmospheric medium: such interplay is still far from a satisfactory knowledge and understanding (Jones and Harrison 2004; Kellogg and Griffin 2006; Deguillaume et al., 2008; Tang, 2009; Bowers et al., 2010). On the other side, bacteria and PBAP dispersed in the atmosphere can be chemically active (Ariya et al., 2002) and favour the formation of ice and cloud condensation nuclei (Ariya et al., 2009; Hoose et al., 2010; Möhler et al., 2008). Primary biological aerosol particles are generally assumed to be efficient CCN, provided that their surfaces are wettable (Després et al., 2012). Bauer et al. (2003) suggested that the chemical composition, structure and hydrophilicity of the surface layer of bacteria could play important roles in CCN activity. Ariya and Amyot (2004) proposed that bio-aerosols have a potential role in the chemistry of organic compounds in the troposphere via microbiological degradation and hence inducing changes in the IN or CCN ability of organics in atmosphere.

So far, PBAP have been studied in-field through a variety of sampling and analysis techniques and addressing their physical, chemical, and biological properties (Reponen et al., 1995; Li and Lin, 1999; Brodie et al., 2007; Georgakopoulos et al., 2009; Fahlgren et al., 2010; Lee et al., 2010; Urbano et al., 2011). The connection between PBAP and dust dispersion and transport over very long distances (Goudie and Middleton, 2006) deserves a particular mention. Dust clouds may contain high concentrations of microbiota, e.g. fungal spores, plant pollen, algae and bacteria. Bio-aerosols associated with dust events can spread pathogens over long distances (Prospero e al., 2005; Griffin, 2007; Nava et al., 2012; Van Leuken et al., 2016) and can impact ecosystem equilibria, human health and yield of agricultural products. For many microorganisms long-range and high-altitude transport in the free atmosphere can be very stressful due to strong ultraviolet radiation, low humidity (inducing desiccation), too low or too high temperatures, and complex atmospheric chemistry (e.g. presence of radicals or other reactive species) (Després et al., 2012; Zhao et al., 2014). Only very resistant organisms are able to survive, so the composition of microbiota can change during the long airborne transport prior deposition (Meola et al., 2015).

Airborne bacterial communities are highly diverse, and variations in their species diversity are quite complex. The bacterial composition in air is strongly dependent on many factors such as seasonality, meteorological factors, anthropogenic influence, variability of bacterial sources and many other variables. Still, the general trend from available reports is that bacteria found in the air often belong to groups that are also common soil bacteria (e.g. Firmicutes, Proteobacteria, Actinobacteria) (Després et al., 2012). Due to their small size, bacteria have a relatively long atmospheric residence time (on the order of several days or more) compared to larger particles and can be transported over long distances (up to thousands of kilometres). Measurements show that mean concentrations in ambient air can be greater than $1 \times 10^4$ cells $m^{-3}$ over land, whereas concentrations over the sea may be lower by a factor of 100-1000 (Burrows et al., 2009a, Burrows et al., 2009b).

Bio-aerosols also seem to play an important role in the reactivity of particulate matter. They can induce Reactive Oxygen Species (ROS) production and modify particulate matter (PM) toxicity due to their ability to modulate the oxidative potential (OP) of toxic chemicals present in PM (Samake et al., 2017).

Therefore, within the bacterial survival studies there are four interconnected topics. One is related to health issues: exposure to bio-aerosols has been linked to various health effects (disease spreading e.g. Meningitis and bioaero-contamination, like

Legionella and refrigerating towers. Pearson et al, 2015; Ghosh et al, 2015; Sala Ferré et al, 2009). Another topic is connected to climate and CCN/IN impact, where viability and proliferation of airborne bacteria are the significant investigation subjects (Bauer e al., 2003; Deguillaume et al., 2008; Amato et al., 2015). A biogeochemical issue is related to the long range transport of bacteria and dust events, since bacteria can stick to dust particles and can be more efficiently (i.e. remaining viable) transported through long distances. (Meola et al., 2015; Nava et al., 2012; Van Leuken et al., 2016).

## 1.2 Atmospheric simulation chambers and bacteria

The study of relevant processes taking place in the Earth atmosphere is usually pursued through a wide range of field observations where complicate, unexpected and interconnected effects are often difficult to disentangle. The possibility of planning and performing experiments in controlled conditions is therefore highly desirable. This need triggered the concept and the development of the atmospheric simulation chambers (ASCs in the following), i.e., small- to large-scale facilities (with volumes ranging between a few to hundreds cubic meters), where atmospheric conditions can be maintained and monitored in real time for periods long enough to mimic the realistic environments and to study interactions among their constituents (Finlayson-Pitts and Pitts, 2000). ASCs have been used to study chemical and photochemical processes that occur in the atmosphere, such as ozone formation (Carter et al., 2005 and references therein) and cloud chemistry (Wagner et al., 2006) or aerosol-cloud interaction (Benz et al., 2005), but the high versatility of these facilities allows for a wider application covering all fields of atmospheric aerosol science. A full list and review of the approach and of the main facilities around the world can be found in Becker (2006). In Europe, there are several ASCs organized through the network EUROCHAMP-2020 (see all the details at the link www.eurochamp.org).

Since the interplay of bio-aerosol and atmospheric conditions is still poorly known, suitable facilities are needed, where transdisciplinary studies gathering atmospheric physics-chemistry and biology issues are possible.

Experiments conducted inside confined artificial environments where physical and chemical conditions/compositions can be controlled, can provide information on bacterial viability, biofilm and spore formation and endotoxin production. Currently, the literature reports several examples of studies performed in small reactors (Levin et al.,1997; Griffiths et al., 2001; Ho et al., 2001; Ribeiro et al., 2013; Sousa et al., 2012). The use of atmospheric simulation chambers has been much more limited and focussed on the interaction of bacteria with atmospheric parameters, regarding bio-aerosols release effects (Jones and Harrison, 2004), and on ice nucleation and cloud condensation (Möhler et al., 2008; Bundke et al., 2010; Chou, 2011).

In 2014, some of the co-Authors of the present work, designed and performed an exploratory experiment (Brotto et al., 2015) at the CESAM (French acronym for: Experimental Multiphasic Atmospheric Simulation Chamber) atmospheric chamber (Wang et al., 2011). On colonies of *Bacillus subtilis* injected, then extracted from CESAM on Petri dishes, they could observe a clear increase of bacterial viability when concentrations of $NO/NO_2$ and $CO_2$ were contemporarily maintained inside the simulation chamber at a level of about 65/630 ppb and 400 ppm, respectively. *Bacillus subtilis* is a well-known Gram positive bacterial strain (Burrows et al., 2009; Gandolfi et al., 2013) and the viability increase observed in the two experiments was by a factor 35 and 10, respectively (Brotto et al., 2015). Such experimental evidence made clear that the effects of atmospheric pollution on bacteria viability could be studied in atmospheric chambers. In order to perform systematic studies to resolve and describe the physical and chemical mechanisms ruling these interactions, dedicated facilities with a microbiology laboratory linked to the ASC for the handling and characterization of bio-aerosol are needed.

Prompted by the outcomes of pilot experiments (Amato et al., 2015; Brotto et al., 2015), a new dedicated atmospheric chamber, ChAMBRe (Chamber for Aerosol Modelling and Bio-aerosol Research), has been designed and installed in Genova (IT). While ChAMBRe, as other ASCs, is a multi-purpose facility, the outcomes of the correlation between bacteria viability and atmospheric condition/composition will provide the input for developing ad-hoc modules to be then implemented in chemical transport models. This can be done following a scheme often used for the chemical mechanisms parameterization (see for example the smog chamber experiments used for the evaluation of Carbon Bond mechanisms in Parikh et al., 2013).

Such software tools, are widely used both in scientific research and in air quality evaluations, to predict the fate (i.e. transport, deposition and chemical changes) of the atmospheric pollutants and, at the moment, they do not include any biological patch.

## 2. Description of the facility

### 2.1 ChAMBRe main structure

ChAMBRe is installed at the ground floor of the building hosting the Department of Physics of the University of Genova, where it is jointly managed by the Italian National Institute of Nuclear Physics (INFN) and the Physics Department (www.labfisa.ge.infn.it). Since the beginning of 2017, ChAMBRe is one of the nodes of the EUROCHAMP-2020 network with specific tasks on bio-aerosol studies.

CHAMBRe has a cylindrical shape with domed bases (Figure 1). It has maximum height and diameter of 2.9 m and 1 m,
respectively and a total volume of 2.23 m$^3$. The latter includes all the secondary volumes connected to the main body and has been determined measuring the volume of air needed to bring the chamber at atmospheric pressure after an evacuation down to 5 x 10$^{-2}$ mbar. The main body is divided into three parts: two domed cylinders (see Figure 1) connected by a central ring 60 cm height. The lower dome has a bottom aperture with a pass through for the shaft of a fan and two lateral ISO-K250 flanges. The central ring allocates symmetrically six flanges (two with a diameter of 40 cm and four with a diameter of 10 cm). Finally,
the top cylinder is equipped with two lateral and symmetrical ISO-K100 flanges plus another flanged aperture (ISO-K250) on the dome. The interior of the chamber can be accessed through the two ISO-K400 flanges or removing the top dome by a crane. One of the two flanges in the bottom part is connected through a pneumatic valve to a smaller horizontal cylinder, (length = 1 m) which hosts a movable shelf designed to move specific samples inside the chamber as described in section 4.3. The lower dome is hold by a metallic support to maintain the entire structure in vertical position (Figure 2).

While ChAMBRe has been designed to operate at atmospheric pressure, the second ISO-K250 flange of the lower cylinder is connected to a composite pumping system (a rotary pump model TRIVAC® D65B, Leybold Vacuum, followed by a root pump model RUVAC WAU 251, Leybold Vacuum) which can evacuate the internal volume to a vacuum level of about 5 x 10$^{-2}$ mbar in about 15 minutes. . A safety valve (Leycon Secuvac DN 63, Oerlikon Leybold Vacuum) is mounted as a gate between the pumping system and ChAMBRE: in the event of a power failure it automatically closes in less than one ms, thus
preventing possible backwashes of the pumps oil inside the chamber. The return to atmospheric pressure is a two-step procedure: first pure N$_2$ from a compressed gas cylinder is flushed in, until a pressure of 5 mbar is reached, and then the ambient air can enter the chamber through an absolute HEPA filter (model: PFIHE842, NW25/40 Inlet/Outlet - 25/55 SCFM, 99.97 % efficient at 0.3 μm) and a zeolite trap (upstream).

### 2.2 Basic equipment

To favour the mixing of the gas and aerosol species in the reactor a fan is installed in the bottom part of the chamber (Figure 1). It is a standard venting system with four metallic arms of 25 cm length each connected to an external engine through a rotating shaft. A particular pass through has been designed and built at INFN-Genova to ensure the vacuum seal. The fan speed can be regulated by an external controller and varied between 0.0 Hz and 50 Hz in steps of 0.1 Hz (0 to 3000 rpm, in steps of 6 rpm).

A set of two pressure gauges is used to measure the atmospheric pressure inside and outside the chamber. A MKS Instruments 910 DualTrans™ transducer is installed inside with a measuring range of 5 x 10$^{-4}$ to 2 x 10$^3$ mbar and an accuracy of ±10 % of its reading, in the range of 5 x 10$^{-4}$ to 1 x 10$^{-3}$ mbar, ±5 % of reading in the range of 10$^{-3}$ to 15 mbar and ±0.75 % of reading in the range of 15 to 1000 mbar. The pressure transducer contains two separate sensor elements: a MicroPirani™ sensor element, based on measurement of thermal conductivity, and a Piezo sensor, based on measurement of the mechanical
deflection of a silicon membrane relative to an integrated reference vacuum. The Piezo measures true absolute pressure

independent of gas composition and concentration. A Vaisala BAROCAP® Barometer PTB110 is installed outside the chamber with a measuring range of $5 \times 10^2$ to $1.1 \ 10^3$ mbar and accuracy of $\pm 0.3$ mbar at 20° C.

Internal temperature and relative humidity are continuously measured by a HMT334 Vaisala® Humicap® humidity and temperature transmitter for high pressure and vacuum application (up to 100 bars). This sensor is mounted in the upper ISO-K100 flange on the top dome. In the operative range (from 15 to 25 °C) the accuracy is $\pm 1$ %RH (0 to 40 %RH) and $\pm 1.7$ %RH (90 to 100 %RH) and $\pm 0.2$°C at 20 °C.

All the atmospheric gauges are connected to a NI Compact-RIO acquisition system (based on the NI cRIO-9064 controller) which also allows the remote monitoring of the ChAMBRe parameters through an Ethernet connection.

Two type of UV lamps are permanently installed inside the chamber. A 90 cm long lamp is inserted through the flange in the top dome (Figure 1): it produces a 85 W UV radiation at $\lambda = 253.7$ nm (UV-STYLO-NX, Light Progress srl) which is used to sterilize, without producing ozone, the chamber volume, in particular after any experiment with bio-aerosol. A second type of lamp, producing UV radiation at $\lambda < 240$ nm, can be inserted through one of the ISO-K100 flanges of the central ring to generate ozone. Two different units of mercury lamps (length = 5 cm, power = 6 W and length = 20 cm, power = 10 W; both of BHK Incorporated, Analamp models), can bring ozone concentration inside ChAMBRe from zero to about 300 ppb in about 30 or 15 minutes, respectively.

### 2.3 Instruments connected to ChAMBRe

The large number of free flanges in the main structure gives the possibility to connect several external instruments to ChAMBRE.

Polydispersed aerosol can be sprayed into the simulation chamber using a Blaumstein Atomizer (BLAM, single-jet model, CH Technologies), connected to the chamber with a curved stainless-steel tube (length = 50 cm, diameter = 1.5 cm). The single jet BLAM is specifically designed to provide bio-aerosols with the enhanced viability of microorganisms for aerobiology research (Zhen et al., 2014) with respect to the Collison nebulizer, employed in the pilot test performed by Brotto et al. (2015). The BLAM's viability is essentially due to its efficiency in that it utilizes minimal energy to properly aerosolize a liquid. The single-jet BLAM is used in one-pass mode, where the liquid medium is subjected to the sonic air jet only one time. The atomizing head is composed of two main parts: Nozzle Body and Expansion Plate. The atomization occurs when the pressurized air (air flow 2 lpm, pressure 3.8 bar) pushes at sonic velocity through a precisely laser cut ruby crystal (fixed size 0.010" diameter) pressed into the Nozzle Body, while the liquid with particles is carried into a cavity between the Nozzle Body and Expansion Plate at a desired flow rate (liquid feed = 0.4 mL min$^{-1}$) using precision pump (NE-300 Just Infusion™ Syringe Pump, New Era Pump Systems, Inc.). The properties of the aerosol generated by the single-jet BLAM are a function of the jet hole size, depth of the liquid cavity and expansion cone size. The atomizer features a modular design, composed of five interchangeable plates which enable it to accommodate liquids of varying properties to produce aerosols in specific size ranges and output concentration, with a nebulization efficiency (i.e. mass ratio between the mass of the produced aerosol to the mass of the solute or of the material suspended in the liquid inserted in the BLAM) between 1 % and 8 %. In this work, the expansion plate with a cavity depth and a cone diameter of 0.001 and 0.020 inch, respectively, has been used. The accelerated air jet breaks up the liquid into droplets. The aerosol generated by this process is sprayed downwards inside the jar where the larger droplets are collected on the liquid surface due to impaction as they cannot make the U-turn while the finest droplets are forced up through the outlet tube on top of the BLAM lid. The result is a very fine mist, well within the respirable range (i.e. with diameter smaller than 10 μm) and with narrow size distribution. The size distribution, immediately after the injection of physiological solution (with or without bacteria) in ChAMBRe, shows a mean value of 0.45 μm with a standard deviation of 0.25μm. This information, however, is just a typical figure since the actual size depend on the solution we nebulize according to the type and concentration of the solute.

Aerosol samplers and multi-stage cascade impactors can be easily connected through the ISO-K flanges and maintained in operation for times depending on their nominal flow and the needs of the particular experiment (e.g. a typical 10 L min$^{-1}$ device, like the 13-stage rotating NanoMoudi-II™ - Nano-Micro orifice uniform deposit impactor, Model 125B, MSP Corporation; Hwan et al., 2010 - extracts a 10 % of the total chamber volume in about 20 minutes). A similar figure holds for impingers (*Flow Impinger* by Aquaria srl) which can be filled with 20 mL of sterile physiological solution. Such devices must be operated at a constant air flow of 12.5 L min$^{-1}$ (e.g. by a Low Capacity Pump Model LCP5, Copley Scientific).

Particle concentration inside the chamber is measured continuously by two different instruments: a Scanning Mobility Particle Sizer (SMPS, GRIMM Technologies, Inc.) and an Optical Particle Counter (OPC, mod. Envirocheck 1.107, GRIMM Technologies, Inc.).

The SMPS is formed by three components in sequence: a neutralizer (i.e. a bipolar diffusion charger) supplied by Eckert & Ziegler Cesio (Prague), a differential mobility analyzer (DMA, model 55-U) and a condensation particle counter (CPC, model 5403), both from Grimm GmbH (Ainring, Germany). The neutralizer is based on a radioactive source of $^{241}$Am with an activity of 3.7 MBq. The DMA is available with two different columns, working alternatively in the size range 5.5-350.4 nm (MDMA), and 11.1-1083.3 nm (LDMA), and classifying particles in 50 dimensional classes. Scanning the voltage through the entire electrical particle mobility range requires about 5 min with MDMA and about 10 min with LDMA. If necessary (relative humidity >80 %), the system is equipped with a dedicated air dryer to be inserted upstream of the DMA. A pre-impactor can be also used to remove particles larger than a fixed upper size limit. In the CPC, downstream of the DMA, the particle size is increased by n-butanol condensation on their surface and then the particles are optically counted. The CPC can also be operated as a standalone unit to measure the total particle concentration, with a response time of 4 s and a sensitivity to particle size larger than 4.5 nm. The maximum measurable concentration can reach $10^7$ particles cm$^{-3}$. Both the CPC and the SMPS are operated at an air flow of 0.3 L min$^{-1}$ at atmospheric pressure. To prevent possible damages, the inlet is connected to ChAMBRe through a gate valve which is closed before any evacuation procedure. The SMPS has been connected to ChAMBRe through a smoothly bended pipe in a way to have an horizontal length of about 10 cm followed by a vertical part of about 30 cm.

The OPC is a Grimm 1.107 - Envirocheck version, which operates in 31 size intervals with diameters in the 0.25-32 µm size range with a 6-sec time resolution. The Grimm OPC uses a dehumidification system which operates when ambient relative humidity is higher than 70 %. This optical particle counter has a patented light scattering technique based on an advanced low water sensitive laser source (λ=675 nm). The OPC is factory calibrated via monodisperse Latex particles for size classification. The reproducibility of the OPC in particle counting is ± 2 % (Putaud et al., 2004). The OPC working flow is 1.2 L min$^{-1}$ and it is connected to ChAMBRe through a gate valve which is closed before emptying the chamber volume. This position allows for sampling directly sucking from one of the large flanges without any connecting tube.

The ozone concentration is monitored by a M400A Ozone Analyzer from API (Advanced Pollution Instrumentation, Inc.). The M400A uses a system based on the Lambert-Beer law for measuring ozone in ambient air. A 254 nm UV light signal is passed through the sample cell where it is absorbed in proportion to the amount of the ozone present. Periodically, a switching valve alternates measurement between the sample stream and a sample that has been scrubbed of ozone. The instrument has a sampling rate of 0.8 L min$^{-1}$, a response time of 6 seconds and a detection limit of 0.6 ppb (update UV Photometric Ozone Analyzer, model O342e from Environnement SA).

The nitrogen oxides (NO and NO$_2$) concentrations are monitored by an AC32e, from Environnement SA. The AC32e utilizes the principle of chemiluminescence, which is the standard method for the measurement of NO and NO$_2$ concentration (EN 1421), for automatically analyzing the NO - NOx and NO$_2$ concentration within a gaseous sample. The analyzer measures the photons emitted after the reaction between NO and O$_3$. The analyzer initially measure the NO concentration in the sample, through NO ozone oxidation. Subsequently, the sample passes through the heated molybdenum converter which reduces NO$_2$ to NO and is then mixed with ozone in the reaction chamber and the resulting NO concentration is determined. in this way,

the signal is proportional to the sum of the molecule NO and $NO_2$ (reduced to NO in the converter) in the sample. With a sampling rate of 0.66 L min[-1] this instrument reaches a detection limit of 0.2 ppb with a response time of 40 s.

## 3. Characterization

### 3.1 Aerosol particle lifetime

Depending on kinetics, processes in the atmosphere have typical reaction times ranging from a few seconds up to several days. For this reason, in the case of simulation chambers, the evaluation of aerosol particle lifetime is of primary importance: it is necessary to keep in suspension enough aerosol for a sufficient time, in order to allow chemical or biological transformations of particles. Aerosol particle lifetime in chambers depends on many factors e.g. wall losses caused by adsorption/deposition, diffusion and mixing processes, gravitational settling, electrostatic drawing, all of them depending of course on particle properties (i.e. density, dimensions, shape and vapour pressure).

For the characterization of particle lifetime in ChAMBRe, the Blaumstein Atomizer (BLAM) was used. By feeding the BLAM with saline solutions (NaCl and $(NH_4)_2SO_4$) with different concentration (up to very concentrated solutions, about 10 g L[-1]), it is possible to generate polydispersed particles with continuous size distributions from few nm up to about 5 µm. During these experiments, the mixing fan was kept on at a constant rotation speed of 5 Hz, this resulting in a mixing time of about 2 min. Thanks to the combined SMPS-OPC measurements, the aerosol particle lifetime was measured as a function of particle size (Figure 3). For each size bin of the two instruments, particle lifetime has been determined by fitting the mass decay curve with a simple first order exponential. Relative Humidity in ChAMBRe during the measurements was around R.H.= 47%. Aerosol dilution due to the air flow through the two counters (in total: 1.6 L min[-1]) was taken into account and properly corrected; the chamber is designed to ensure that the pressure is kept constant: the same amount of clean air is introduced into the chamber through the input from the HEPA filter. The first time interval after each injection, when coagulation could take place, was excluded in the analysis, considering this way the concentration values smaller than $10^4$ particle cm[-3] only. Results are reasonable and very close to literature values (Lai and Nazaroff, 2000; Cocker et al. 2001; Wang et al., 2011); in particular experimental data are nicely reproduced (i.e. the mean discrepancy between measured and calculated values is around 50%) by the wall deposition model described in Lai and Nazaroff, (2000) treating ChAMBRe as a rectangular cavity with a friction velocity of ca 6 cm s[-1] (Figure 3). Particle lifetime in ChAMBRe varies from few hours to about 1 day depending on particle size. The uncertainty on particle life-time plotted in Figure 3 has been evaluated on a pure statistical basis. Actually, in the size region between 300 and 600 nm, both the SMPS and OPC data could be particularly sensitive to other effects (e.g. background fluctuation for the SMPS, systematic artifacts in the first OPC bins) which have not been fully investigated in this work and that do not change the typical feature depicted in Figure 3. While nominal particle lifetimes are important parameters to design the experiments in the chamber and their typical time window, their values can actually vary according to the specific characteristics of the injected/formed particles.

### 3.2 Ozone and wall reactivity

The presence of walls obviously influences the chemical and physical dynamics of the experiments carried out inside simulation chambers, as the gaseous species can be lost to the chamber walls. To describe the behavior of the walls of our chamber, we considered the dark reactivity of ozone, due to its chemical reactivity towards surfaces, its relevance to chamber experiments (as reactant or as sterilization agent) and as atmospheric oxidant.

A series of five experiments have been done with initial concentration ranging from 300 to 1000 ppbv. The ozone concentration in the chamber was monitored as a function of time. The pseudo-first order rate for loss processes is equal to $(3.04 \pm 0.40) \times 10^{-5}$ s[-1] and it is in good agreement with what reported in the literature for other similar facilities (Wang et al., 2011). This parameter is highly dependent on the chamber wall material, on its history, related to the cleaning protocol and the operating

conditions such as temperature or relative humidity (Wang et al., 2011). As a consequence, the quantification must be carried on regularly and before each set of experiments for any type of study.

## 3.3 Background levels (PM, O₃, NOx)

The background level of particles inside the chamber was measured by SMPS and OPC. The coupling of the two counters provides a comprehensive picture of the particles inside the chamber ranging from few nm up to 31 microns (for more information, see section 2.3). After each experiment, the chamber is cleaned by a multi-step procedure: the UV lamp (see sec. 2.1) is first switched on for 10 min, the chamber is then evacuated and vented to atmospheric pressure through an HEPA filter (section 2.1). Afterwards, a high ozone concentration (>500 ppb) is produced to be sure to sterilize any part of the set-up possibly not reached before by the UV rays. Finally, the chamber is evacuated and vented again.

Background level measurements performed subsequently to chamber cleaning showed no significant particles presence (i.e. about 2 and 0.5 particle cm$^{-3}$, respectively in the SMPS-LDMA and OPC range).

Background concentrations of O₃ and NO$_x$, could be introduced in the chamber during the venting after an evacuation, since both the gases can be present in the room air: concentration values measured periodically in the chamber along 4 months turned out to be smaller than 1-2 ppb i.e. close to the analyser sensitivity (see section 2.3).

## 4. Protocols to prepare, inject, expose and collect bacteria

The usefulness of ASCs in providing new possibilities for the study of bacteria and other biological particles in air critically depends on the associated protocols, which are essential to understand how the bacteria survive and if they are in able to grow and reproduce in the atmospheric conditions of the simulation chamber. In this section we describe the standard methodology developed for the bio-aerosol experiments (injection, collection and storage) and the related experimental conditions, that should be representative of the typical environmental ones.

## 4.1 Bacterial strains

Experimental procedures involved two strains consisting of *Bacillus subtilis* (ATCC® 6633™) and *Escherichia coli* (ATCC® 25922™). These microorganisms are extensively used as model organisms in microbiology and molecular biology fundamental and applied studies (Lee et al., 2002).

*Bacillus subtilis* is a Gram-positive, rod-shaped bacterium with length ranging between 2.5 and 6.5 µm. It is commonly found in soils but has been also observed in other environmental matrices such as water and air (Earl et al., 2008). It has a wide commercial use as it is nonpathogenic. *B. subtilis* serves as a model organism and is considered a reference for cell differentiation and adaptation. This model status makes it one of the most extensively studied organisms in nature given its ability to survive and even thrive in a wide range of harsh environments (Earl et al., 2008).

*Escherichia coli* is a Gram-negative, rod-shaped, enterobacter, is about 1–2 µm long and about 0.25 µm in diameter. It is a common inhabitant of the gastrointestinal tract of warm-blooded animals, including humans, but recent studies have reported that some specific strains of *E. coli* can also survive for long periods of time, and potentially reproduce, in extra-intestinal environments. *Escherichia coli* is among one of the most studied model organism. Its fast-growth characteristics under optimal conditions make it suitable as host organism for many gene manipulation systems, producing countless enzymes and other industrial products, and to study the evolution of microorganisms (Jang et al., 2017).

## 4.2 Preparation of bacterial suspension and injection in ChAMBRe

Several techniques for bacteria and bio-aerosol characterization are available on site. In the same building that hosts the atmospheric simulation chamber there is a basic microbiology lab equipment allowing for culture analysis in vitro (isolation,

identification, growth) and biochemical tests (e.g. catalase and oxidase): autoclave (Asal mod.760), vortex, centrifuge and microcentrifuge (Eppendorf centrifuge 5417R), water purification system Milli-Q (Millipore-Elix), incubator for temperature control Ecocell and Friocell MMM Group, Steril-VBH Compact "microbiological safety" cabinet, Thermo electron corporation steri-cycle HEPA Class 100 incubator; optical microscope (Nikon Eclipse TE300) for bacterial detection and live/dead discrimination by epifluorescence with specific dyes and for immunoassay fluorescence to label antigenic bacterial target, fluorescent molecule or enzyme. The transfer of bacteria from the biological laboratory to the simulation chamber takes only a few minutes, ensuring a quickly execution of the chamber experiments, once the desired phase of bacteria growth is reached, and then a quick treatment of the samples collected after the experiments in the room.

The same culture preparation technique was applied at both the bacterial strains, in order to minimize experimental variations. Firstly, it is important to ensure the maximum bacteria cells viability prior to the injection. Typically, to understand and define the growth of a particular microbial isolate, cells are placed in a culture medium in which the nutrients and environmental conditions are controlled. If the medium provides all nutrients required for growth and environmental parameters are optimal, a growth curve can be obtained by measuring the increase in bacterial number or mass as a function of time. Different distinct growth phases can be observed within a growth curve: these include the lag phase, the log phase, the stationary phase, and the death phase. Each of these phases represents a distinct period of growth that is associated with typical physiological changes in the cell culture. Therefore, the growth curve for both of bacterial strains was obtained quantifying the rate of change in the number of cells in a culture per unit time thus identifying the mid-exponential phase (log phase), where the maximum viability of the cells is ensured and the number of dead microorganisms is minimum. *B. subtilis* was purchased as water soluble freeze-dried Selectrol discs. The discs were dissolved in sterile Tryptic Soy Broth (TSB), also known as soybean-casein digest medium (SCDM), incubated at 37° C for 1 day and then rejuvenated; *E. coli* cells were scrapped off agar medium using sterile plastic loops and suspended in sterile culture broth medium. In both cases, the growth curve was then followed, once every hour, with a spectrophotometer V-530 UV-vis (Jasco International Co. Ltd, Hachioji, Japan), where the number of cells per mL of culture was estimated from the turbidity of the culture. The optical density (OD) of the bacterial solution, measured at a wavelength of 600 nm, is a common method for estimating the concentration of bacterial cells in a liquid. The amount of the light scattered by the microorganisms suspension is an indication of the biomass contents (Sutton, S. 2011). Data, obtained from spectrophotometric measurements ($OD_{600nm}$), were used to estimate when the mid-exponential phase (corresponding an $OD_{600nm}$ of 0.5) is reached. Actually, the number of cultivable cells was counted as Colony Forming Units (CFU), by standard dilution plating: 100 μL of six fold serial dilutions of the solution was spread on an agar non-selective culture medium, and incubated at 37° C for 24 h before counting the formed colonies. Data, obtained from CFU counting on Petri dishes, were averaged and used to estimate the uncertainty range of the bacterial concentration in the solution. The growth curves for the two strains are reported in Figure 4. The measured $OD_{600nm}$ values were fitted with a three-parameter sigmoidal curve (Eq. 1), where Abs is the absorbance, or optical density, measured at 600 nm, *a* and *b* are constants (B. subtilis curve, a is 1.1 ± 0.01, b is 38 ± 2; E. coli curve, a is 0.83 ± 0.01 and b is 41 ± 1).

$$Abs = \frac{a}{1+e^{-((t-t_0)/b)}} \tag{1}$$

Before each injection we followed the bacterial growth up to the mid-exponential phase, reached in about 4 h, thus allowing the bacteria to enter the exponential phase of growth.

Spectrophotometer measurements were used to achieve the correct dilution and also to provide the first evaluation of bacterial concentration in the solution which has to be nebulized, as explained below. The suspension was then centrifuged at 3000 rpm for 10 min, the supernatant was discarded and the pellet was evenly vortexed for 1 min in physiological solution (NaCl 0.9 %) before the injection. The cultivable cell concentration was determined following the above-mentioned procedure. The average on CFU counting is used to estimate the uncertainty range of the bacterial concentration in the nebulized solution.

In each experiment, a volume of 10 mL of the cells suspension, with a concentration of approximately $10^7$ CFU mL$^{-1}$ for *B. subtilis* (OD$_{600nm}$ around 0.5, single values are reported in Table 2) and $10^6$ CFU mL$^{-1}$ for *E. coli,* was prepared for nebulization and placed into a syringe. In particular, for *E. coli*, to obtain the final concentration of $10^6$ CFU mL$^{-1}$, the initial cells suspension with an OD$_{600nm}$ around 0.6 (single values are reported in Table 4) was diluted (1:10, 1:15, 1:20, 1:40) before the injection, to avoid an excessive bacterial concentration on the Petri dishes exposed inside the Chamber (see the paragraph 5.2).

In each experiment, a volume of about 2 or 3 mL of the cells suspension was sprayed into the simulation chamber using the Blaumstein Atomizer (BLAM) described in Section 2.3.

## 4.3 Collection and extraction methods

The main body of ChAMBRe is connected through a ISO-KF250 pneumatic valve to a cylindrical horizontal volume which is accessible from a second ISO-KF250 gate valve (see Figures 1 and 2). The two gate valves completely separate the cylinder, which can be connected to the main chamber or alternatively opened without perturbing the ChAMBRe atmosphere. This home-made device has been specifically developed to ensure the insertion and extraction of bio-aerosol samplers, in order to minimize the risk of contamination. This volume can be evacuated through a by-pass to the ChAMBRe main pumping system and can be then refilled to atmospheric pressure both with particle free dry air or through a pipe connected to the ChAMBRe main body. Inside the cylinder, there is a sliding tray which can be inserted in ChAMBRe by a home-made external manual control (Figure 2) The tray can host up to six Petri dishes (diameter 10 cm, each) which can be inserted in ChAMBRe to collect bacteria (or in general BPAP) directly by deposition onto a proper culture medium. The procedure to insert the Petri dishes in ChAMBRe is organized in consecutive steps (reference to Figure 1 for the valves names):

   a)  With V1 closed, the V2 valve is opened to allow the positioning of the Petri dishes (pre-filled with a suitable amount of culture medium) on the sliding tray

   b)  Valve V2 is closed and the volume inside the pipe is flushed with clean air coming from the chamber.

   c)  The atmospheric pressure inside the pipe is recovered by opening the connection to ChAMBRe

   d)  V1 is opened and the sliding tray is completely inserted in ChAMBRe

   e)  The sterilizing UV lamp (ozone free, see section 2.2) is switched on for 15 minutes to guarantee the Petri dishes sterilization

   f)  The UV lamp is switched off and ChAMBRe is ready for injection of bacteria.

The chamber sterility before the injection of bacteria was tested through a blank experiment by injecting only sterile physiological solution: no bacterial contamination was observed in the four Petri dishes positioned on the sliding tray.

In a standard experiment, once the bacteria have been injected into ChAMBRe, the Petri dishes remain exposed for the desired time and then the sliding tray can be moved back to the pipe. The ventilation system is on during the exposure period, to maintain a homogeneous distribution of particles inside the chamber volume. Closing V1 and opening V2 the Petri dishes can be removed without perturbing the conditions inside the main chamber. The gravitational settling method has been developed to minimize microbial damage, and has been previously proven to be a very suitable way to collect and count viable bacteria colonies (Brotto et al., 2015). After exposure to the chamber atmosphere, Petri dishes are incubated for 24 h at 37° C, after which the number of formed colonies can be counted. It is assumed that the living microorganisms present in the aerosol are deposited on the petri dishes by gravity without undergoing any stress, from those related to the permanence in the experimental setup atmospheric conditions. In this way, it can be assumed that the number of units forming colonies counted on a Petri dish is proportional to the number of aerosolized and suspended living microorganisms within the chamber and also to the concentration value of viable bacteria in the aerosol.

Lee et al., 2002 suggest that the average aerodynamic diameters of generated *E. coli* and *B. subtilis* aerosols were 0.63 and 0.75 µm respectively. If compare these data with data obtained with NaCl solution to determine particles life time in chamber,

the bacteria life time is aspect to be around five hours. The mean global residence time calculated by Burrows et al., 2009b, lie between 2 and 15 days for bacteria traces.

Bacteria from the original liquid suspensions, both in broth and in physiological solution (Section 4.2), were also collected on polycarbonate filters (Isopore membrane track-etched filters, pore size 0.05 µm) with a smooth surface, ideal to study the morphology of cells and possible bacteria aggregates (e.g. biofilm formation) by scanning electron microscopy (Capannelli et al., 2011). The sampling was performed by exposing filters to the stream of aerosols coming out of the nebulizer, through a secondary port connected to the chamber. For electron microscopy observation the simple protocol adopted here is the following. Bacterial suspensions (1 mL) were dehydrated and diluted progressively in a graded series of ethanol bathes (30, 50, 70 and 90 %). This protocol was established by simplifying the standard method named "air drying" (Robinson et al., 1987; Janecek and Kral, 2016), as it was ascertained that the structures of the cells were preserved without requiring the fixation step. Other final treatments (e.g. with tetramethylsilane) were also suppressed as the study of cell ultrastructures were not done in this case as the study. Compared with the original suspensions the final dilution is 1:1000, in order to reach on the filter an optimal surface density, able to maintain the biological particles well separated. Following this step the diluted liquid samples were passed through polycarbonate filters held inside a dedicated filter unit (Swinnex 13 mm filter holder, Millipore Corporation). For each sample, 150 µL were loaded with a micropipette onto the filter in the unit, then a syringe was attached to the upper part of the filter holder, in order to filter the sample by pushing gently the plunger. Then the filter was removed and allowed to dry for 3 hours. Dry filters were cut in half, mounted on Aluminum stubs and sputter coated with carbon before observation by a Field Emission Scanning Electron Microscope (FESEM) Zeiss Supra 40 VP. The selected conditions were: voltage 10 kV, signal in-lens, magnifications ranging from 5000 to 200000×.

## 5. First Experiments

Experiments to study the correlation between bacterial viability and the atmospheric composition/conditions in ChAMBRe rely on an assessed protocol to inject and extract bacteria from the chamber. A first set of experiments was therefore devoted to measuring the reproducibility of the whole process with a clean atmosphere (i.e. with the background levels given in section 3.3) inside ChAMBRe.

### 5.1 Experiments with B. subtilis

Five different experiments were performed in the period from July and November 2017. The protocol described in section 4 was followed for the bacteria growth, the injection in the chamber and the bacteria collection by four Petri dishes inserted by the sliding tray (section 4.3). Values of the atmospheric parameters in ChAMBRe during each experiment are reported in Table 1. The bacteria concentrations measured in the aerosolized solution and the average number of colonies counted on the Petri dishes after the exposure in ChAMBRe are reported in Table 2. The volume of the bacterial suspension injected through the BLAM atomizer was equal to 2 mL, except during the fourth experiment where the volume was increased to 3 mL (Table 2). This ensured that the concentration of viable bacteria injected in the chamber was comparable to the values typical of the real atmosphere (Bauer et al., 2003; Burrows et al., 2009). Taking into account the BLAM nebulization efficiency (section 4.2), the initial aerosol concentration of living microorganisms in ChAMBRe after the injection, was estimated to be around $10^5$ CFU m$^{-3}$. In Table 2, the uncertainties quoted on both injected and collected bacteria are just those deriving from the Poisson fluctuation (i.e. the square root of the number of colonies counted in the Petri dishes) and they do not include any other systematic or statistical contributions. In particular, for the collected CFU, the values reported in Table 2 are the average of the counts of the four Petri dishes exposed in each experiment and that, in each group of four, turned out to be statistically compatible (i.e. within the interval delimited by the statistical uncertainty, the counts in the four petri dishes were in agreement). Despite these simple assumptions, a good correlation between the number of injected and collected CFU was

obtained as shown in Figure 5. Furthermore, the uncertainty on the slope of the correlation curve turned out to be lower than 10 %. This level of reproducibility appears to be adequate to design experiments with different atmospheric conditions (i.e. level of particular pollutants), particularly when compared to the pilot test by Brotto et al. (2015), when much larger variations in the bacteria viability had been observed (see section 1.3). No sizeable effect related to the R.H. in ChAMBRe was observed (crf. Results of Exp. 4 and 5 in Table 2).

## 5.2 Experiments with E. coli

Five different experiments were performed in the period from January and March 2018, following the protocol described in section 4. The values of the atmospheric parameters in ChAMBRe are reported in Table 3. In this set of experiments the relative humidity inside the chamber was increased up to 70 %, compared to the environmental value recorded in the laboratory, by changing the working condition of the humidifier (Benbough, 1967; Cox, 1966; Dunklin and Puck, 1947). *Escherichia coli*, a gram negative bacterium, is more sensitive to the atmospheric conditions inside the chamber than *Bacillus subtilis*, a gram positive strain. As a matter of fact, no CFUs were collected on the petri dishes positioned inside the chamber when the injection of this strain was performed at low relative humidity (RH 35 %, T 20° C). Furthermore, another experiment showed that injecting 2 mL of a cell suspension (concentration of approximately $10^7$ CFU mL$^{-1}$ in physiological solution, RH ~ 70 %) resulted in a huge, uncountable amount of CFUs on the petri dishes, and suggested that a dilution before the injection was necessary.

The dilution factor, the bacterial concentrations measured in the aerosolized solutions and the average number of colonies counted on the Petri dishes after the exposure in ChAMBRe are reported in Table 4. It is worth noting that in the experiments discussed in section 5.1, a narrow interval of $OD_{600nm}$ values, around 0.5, was explored, while in the experiments with *E. coli*, depending on the dilution factor, a larger interval of $OD_{600nm}$ values was spanned.

The volume of the bacterial suspension injected through the BLAM atomizer was equal to 2 mL in the first four experiments and was increased to 2.8 mL in the fifth experiment (Table 4). Figure 6 shows the correlation between the number of injected and collected CFU (left panel), indicating that the uncertainty on the slope of the correlation curve (about 4 %) was even better than the same uncertainty related to *B. subtilis* (about 7 %, Figure 5). In Figure 6, the good correlation between the relative optical density of the cell suspensions and the collected CFU (right panel) is also shown For E. coli suspension, the evaluation of the microbial concentration through the fast and simpler control of the optical density, seems possibly be accurate enough to perform controlled experiments, provided an adequate calibration of the whole procedure is carried out.

Although for this bacterial strain a less concentrated solution was injected, more CFUs were collected on the Petri dishes placed inside the chamber. This result could depend on the fact that the humidity in the chamber was generally greater in the second set of experiments providing to Gram-negative microorganisms a more comfortable environment, but also it could depend on the behavior of the two different bacteria strains.

The FESEM micrographs (Figures 7 and 8) of the bacteria contained in the liquid suspensions before injection (see section 4.3) clearly show that the cells of *B. subtilis* tend to aggregate, forming long chains (Figure 7, left panel), while the cells of *E. coli* are mainly present as single individuals (Figure 8, left panel). Therefore, in the first case it is quite possible that the colonies counted on the Petri dishes originated from a group of cells, while in the second case each colony results presumably from a single viable microorganism.

## 6. Conclusions

A new atmospheric simulation chamber, ChAMBRe, has been installed at INFN-Genova. The facility has been designed to perform experimental studies on primary biological aerosol particles and bacteria in particular. The performance of the new chamber, which may impact on the future experiments on bio-aerosol (i.e. wall reactivity, aerosol particle lifetime, background

levels), has been quantitatively assessed. Furthermore, a protocol to handle the injection and extraction phases has been

thoroughly tested both with Gram positive and Gram negative bacterial strains. With a clean atmosphere maintained inside ChAMBRe, the ratio between injected and extracted viable bacteria turned out to be reproducible at a 10 % level. Such result is the first methodologic step in view of a forthcoming systematic study of the correlation between bacterial viability and pollution levels. Resident times of viable bacteria in ChAMBRe are less than 5 hours, much shorter than the generic residence time in the open atmosphere. However, previous literature studies (Brotto et al., 2015) suggest that such time window is long

enough to observe the effects (i.e. viability change) of bacteria exposure to air pollutants. The assessment of such effects is objective of the fore coming studies at ChAMBRe.

## 7. Competing interests

The authors declare that they do not have any conflict of interest.

## 8. Authors contribution

DM, PB, FP and PP designed and built ChAMBRE; DM, SGD and PP ran all the injections with bacteria; SGD, EG, ADC and LV took care of all the biological issues and measurements, AC, CC, LN and MO performed the SMPS measurements and the FESEM analyses; DM, SGD, CC, MO, JFD, PF and PP performed the measurements to assess the aerosol life time in ChAMBRe and the wall reactivity; FF designed and implemented the acquisition software; JFD and PF provided several advises from their longstanding expertise in the field; DM, SGD, CC, EG and PP prepared the article with the contribute of all

the other authors.

## 9. Acknowledgements

This project/work has received funding from the European Union's Horizon 2020 research and innovation program through the EUROCHAMP-2020 Infrastructure Activity under grant agreement No 730997. The Authors are indebted with the technical staff of INFN-Genova for the intense and talented electro-mechanical works. JFD wish to thank the Physics

department of the University of Genova for granting scientific invitations that have allowed its participation to this work. The authors wish to thank Dr. Houssni Lamkadam and Dr. Claudia Di Biagio (LISA) for the Lai and Nazaroff calculations.

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

**FIGURE CAPTIONS**

**Figure 1: ChAMBRe layout.**

**Figure 2: Left panel: the main structure of ChAMBRe. Right panel: the cylindrical volume (top) which hosts the sliding tray (bottom) used to introduce up to six Petri dishes (or other objects) inside the main ChAMBRe body**

**Figure 3: Particle loss coefficient (β) and life time (secondary vertical axis), versus aerosol size measured in ChAMBRe by NaCl salt injection (21 °C, 47 %RH). The curve resulting from the Lai and Nazaroff 2000 model is also shown for reference (see text). Error bars include statistical uncertainties only.**

**Figure 4: Typical grow curve for *Bacillus Subtilis* (black line, circle) and *Escherichia coli* (red line, triangle): optical density (OD 600 nm) and the corresponding bacteria concentration (CFU/ml) are plotted versus time. Data of concentration curves are reported until the stationary phase.**

**Figure 5: Correlation curve between the number of *B. Subtilis* bacteria injected in ChAMBRe (in units of $10^7$ CFU) and the average count on the four Petri dishes exposed in each experiment.**

**Figure 6: Correlation curve of the average count on the four Petri dishes exposed in each experiment with the number of *E. coli* bacteria injected in ChAMBRe (in units of $10^7$ CFU, left panel) and with the optical density (OD 600 nm, right panel).**

**Figure 7: Detail of *Bacillus subtilis* in physiological solution, magnifications 2000× in the left panel and 100000× in the right panel.**

**Figure 8: Detail of *Escherichia coli* in physiological solution, magnifications 2000× in the left panel and 100000× in the right panel.**

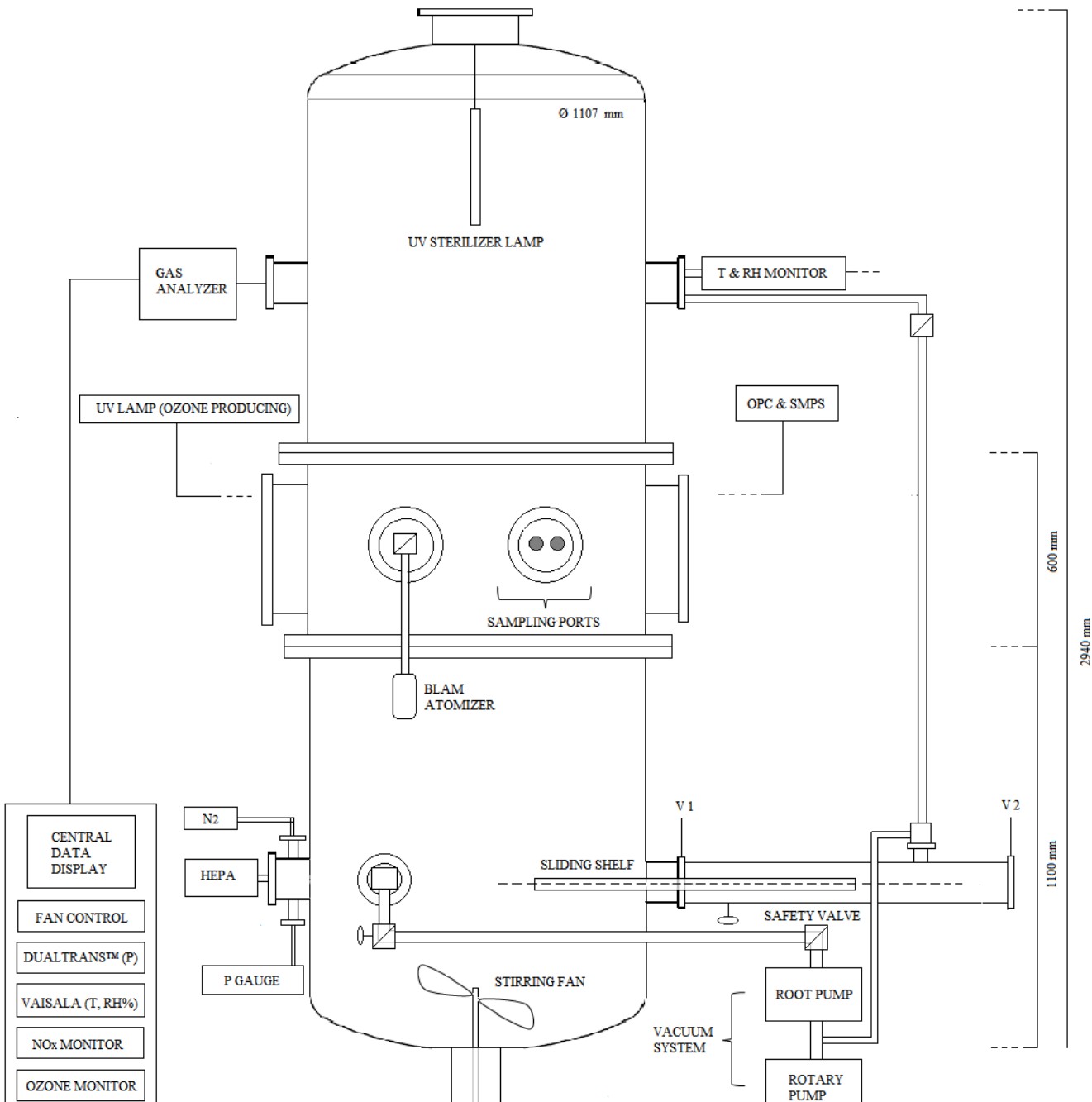


Fig. 1

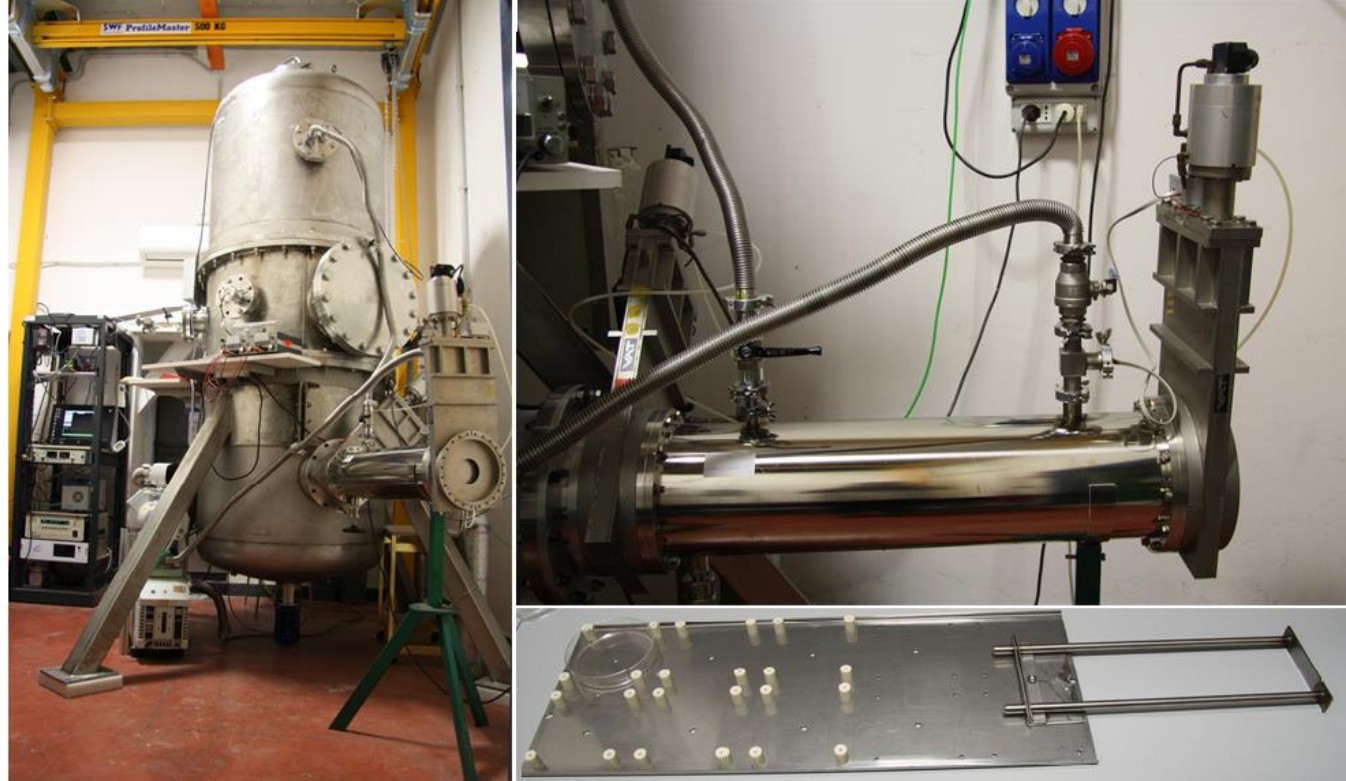


Fig. 2

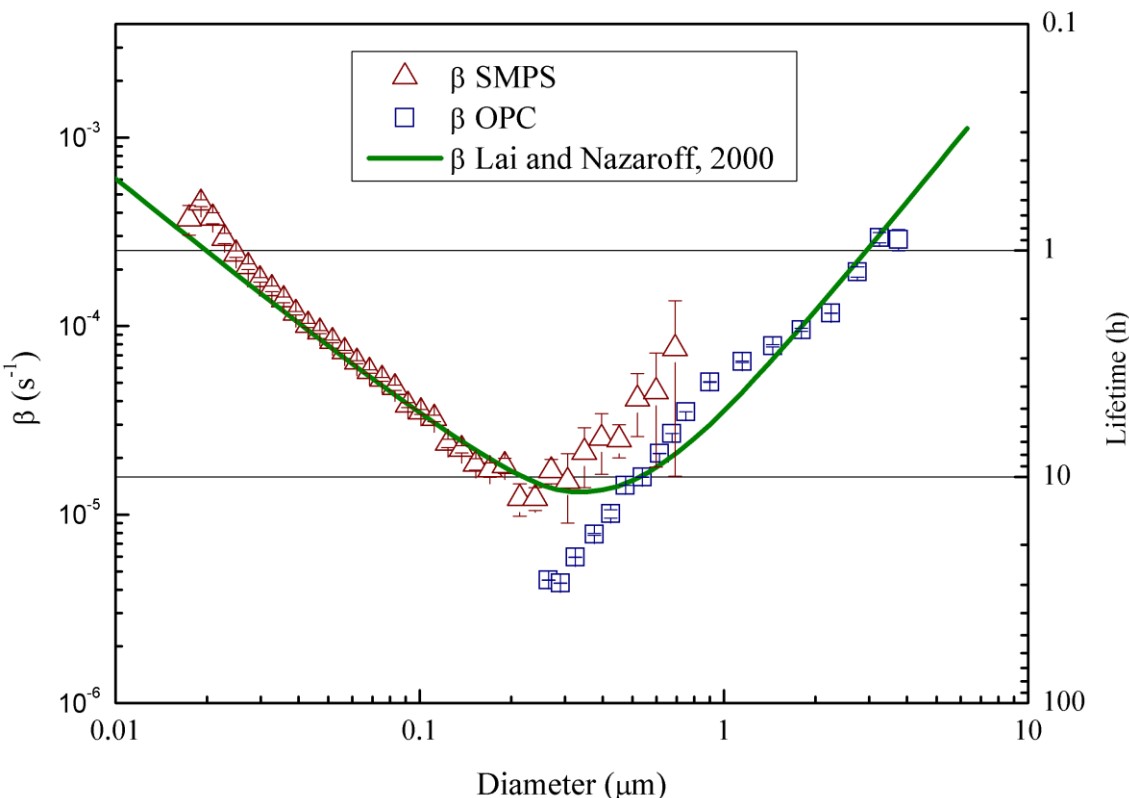

Fig. 3


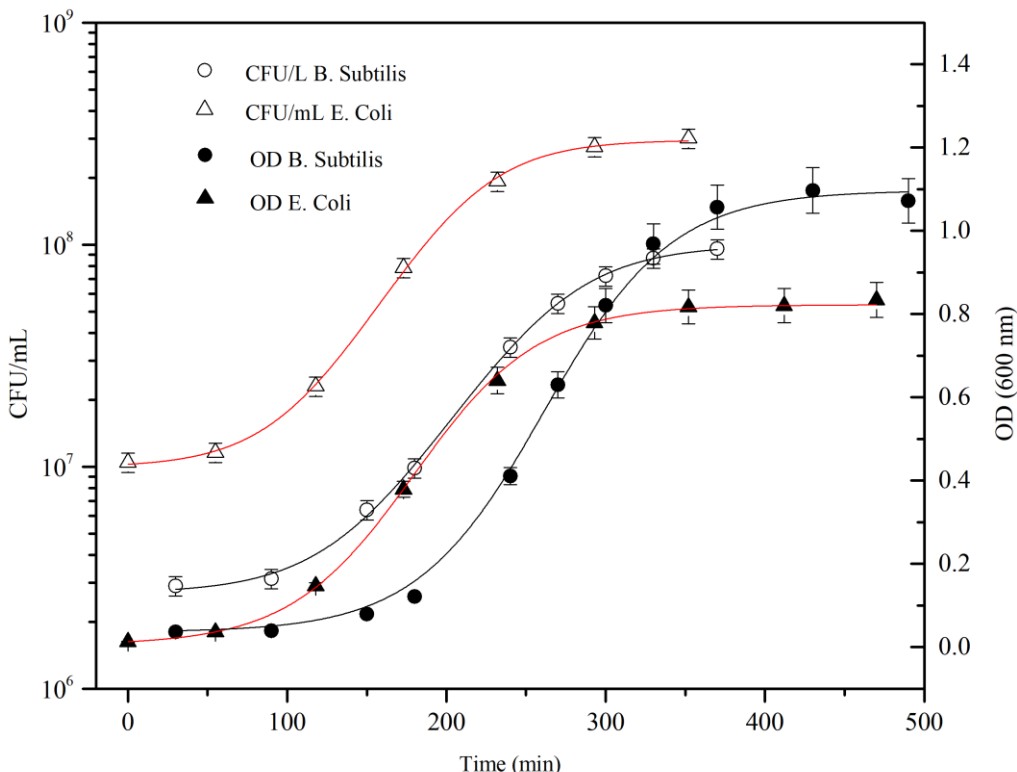

Fig. 4

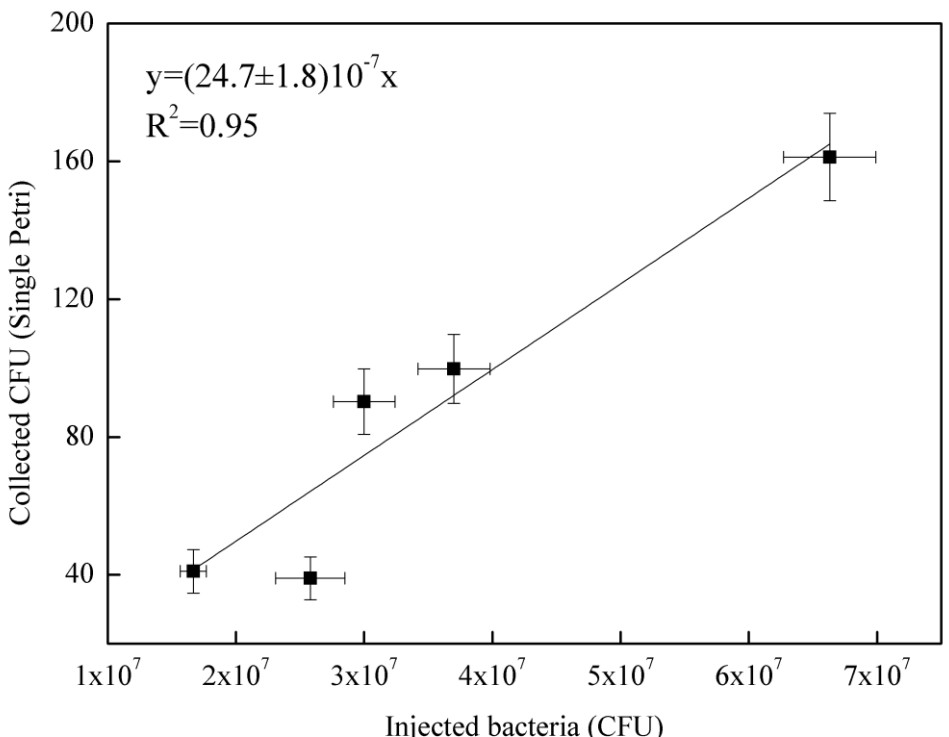

Fig. 5


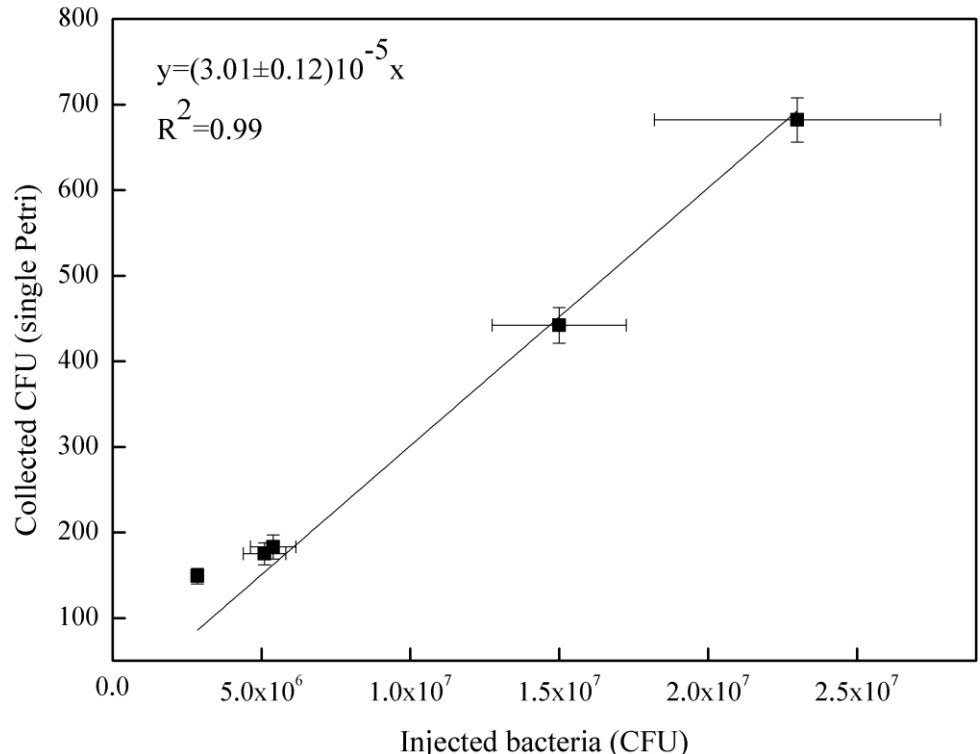

Fig. 6a

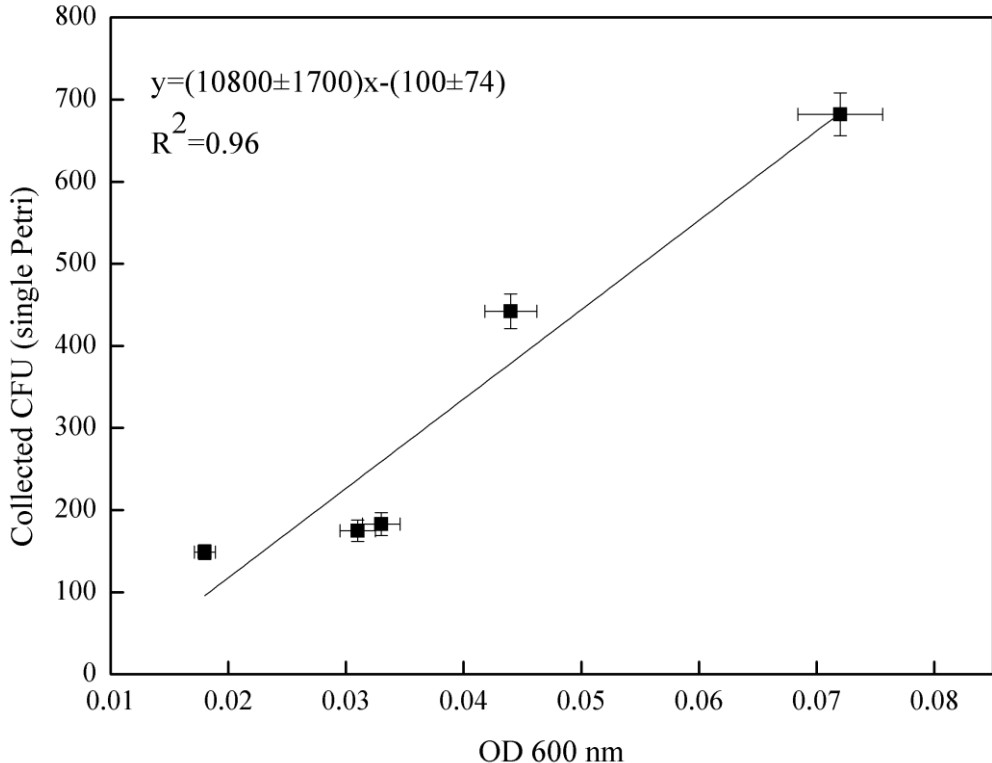


Fig. 6b


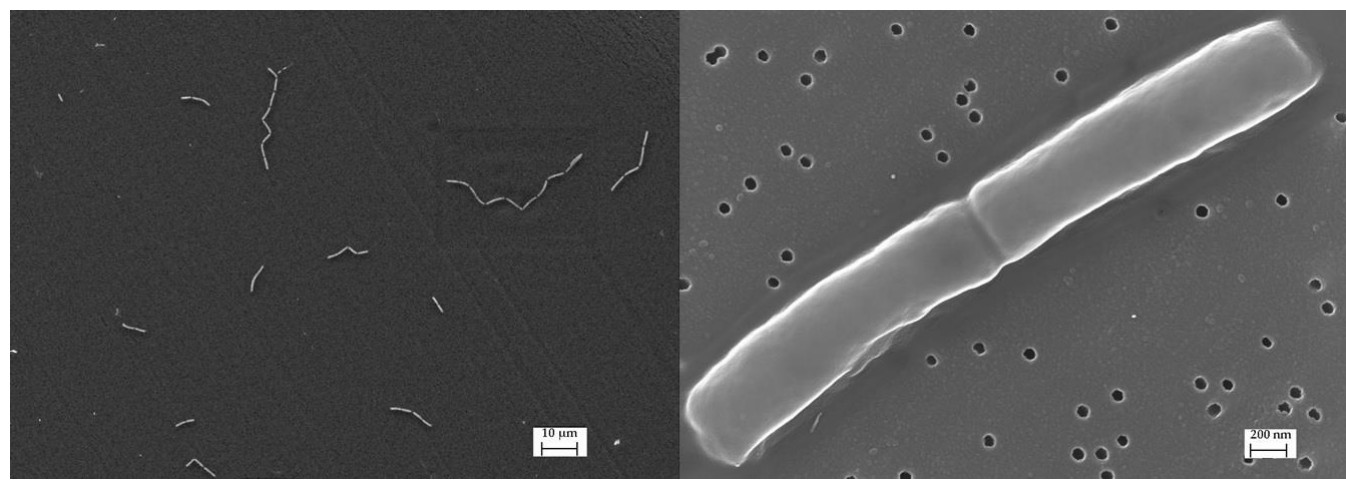

Fig. 7


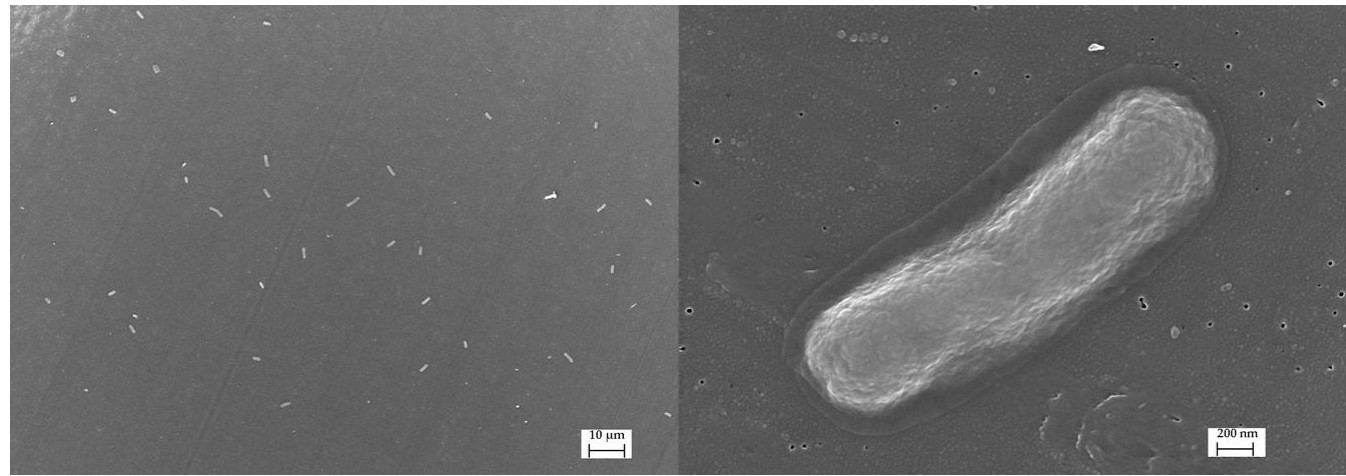

Fig. 8

**Table 1: Environmental parameters (R.H., T, P) in ChAMBRe during the experiments with *B.subtilis*.**

|  | Relative humidity range (%) | Temperature range (°C) | Pressure range (mbar) | Petri dishes Exposure time (hh:mm) |
|---|---|---|---|---|
| Exp. 1 | 55-85 | 22.0-21.1 | 1015-1012 | 05:00 |
| Exp. 2 | 44-71 | 23.7-24.5 | 1010 | 05:20 |
| Exp. 3 | 50-43 | 23.2-21.3 | 1014-1015 | 05:15 |
| Exp. 4 | 44-70 | 22.0-22.5 | 1016 | 05:05 |
| Exp. 5 | 75-79 | 20.1-20.8 | 1005-1007 | 05:00 |


**Table 2: Bacteria concentration (*B. subtilis*) in the aerosolized solution and average number of colonies counted on one Petri dish.**

|  | $OD_{600}$ | Suspension concentration (CFU mL$^{-1}$) x $10^7$ | Bacteria injected CFU x $10^7$ | Average CFU collected |
|---|---|---|---|---|
| Exp. 1 | $0.57 \pm 0.03$ | $1.85 \pm 0.14$ | $3.70 \pm 0.28$ | $100 \pm 10$ |
| Exp. 2 | $0.58 \pm 0.03$ | $3.32 \pm 0.18$ | $6.63 \pm 0.36$ | $161 \pm 13$ |
| Exp. 3 | $0.58 \pm 0.03$ | $1.50 \pm 0.12$ | $3.00 \pm 0.24$ | $90 \pm 10$ |
| Exp. 4 | $0.50 \pm 0.03$ | $0.86 \pm 0.09$ | $2.58 \pm 0.27$ | $39 \pm 6$ |
| Exp. 5 | $0.40 \pm 0.02$ | $0.83 \pm 0.05$ | $1.67 \pm 0.10$ | $41 \pm 6$ |

**Table 3: Environmental parameters (R.H., T, P) in ChAMBRe during the experiments with *E. coli*.**

|  | Relative humidity range (%) | Temperature range (°C) | Pressure range (mbar) | Petri dishes Exposure time (hh:mm) |
|---|---|---|---|---|
| Exp. 1 | 75-77 | 15.8-18.7 | 994 | 05:00 |
| Exp. 2 | 73-77 | 23.1-23.6 | 992-999 | 05:00 |
| Exp. 3 | 78-80 | 19.0-19.3 | 1010 | 05:05 |
| Exp. 4 | 76-83 | 18.6-19.0 | 1007-1009 | 05:00 |
| Exp. 5 | 72-80 | 19.8-20.0 | 1002-1003 | 06:05 |

**Table 4: Bacteria concentration (*E. coli*) in the aerosolized solution and average number of colonies counted on one Petri dish.**

|  | $OD_{600}$ (before dilution) | Dilution factor | $OD_{600}$ (after dilution) | Suspension concentration (CFU mL$^{-1}$) x $10^6$ | Bacteria injected CFU x $10^6$ | Average CFU collected |
|---|---|---|---|---|---|---|
| Exp. 1 | $0.57 \pm 0.03$ | 1:20 | $0.031 \pm 0.002$ | $2.55 \pm 0.36$ | $5.10 \pm 0.71$ | $175 \pm 13$ |
| Exp. 2 | $0.64 \pm 0.03$ | 1:10 | $0.072 \pm 0.004$ | $11.5 \pm 2.40$ | $23.0 \pm 4.8$ | $682 \pm 26$ |
| Exp. 3 | $0.60 \pm 0.03$ | 1:20 | $0.033 \pm 0.002$ | $2.70 \pm 0.38$ | $5.39 \pm 0.76$ | $183 \pm 14$ |
| Exp. 4 | $0.65 \pm 0.03$ | 1:15 | $0.044 \pm 0.002$ | $7.49 \pm 1.12$ | $15.0 \pm 2.25$ | $442 \pm 21$ |
| Exp. 5 | $0.66 \pm 0.03$ | 1:40 | $0.018 \pm 0.001$ | $1.02 \pm 0.07$ | $2.85 \pm 0.20$ | $149 \pm 9$ |