# Peer review of "ChAMBRe: a new atmospheric simulation Chamber for Aerosol Modelling and Bio-aerosol Research"

_Atmospheric Measurement Techniques, 2018_

## Referee Comment (RC1) · Anonymous Referee #1 · 16 Jul 2018

Review of the manuscript amt-2018-147 with title: "ChAMBRe: a new atmospheric simulation Chamber for Aerosol Modelling and Bio-aerosol Research" by Massabò et al.

General comments:

This manuscript describes a new atmospheric simulation chamber and its potential use to simulate the interaction of trace gases and biological aerosol particles like bacteria. Therefore, it fits well in the scope of the journal of atmospheric measurement techniques and focusses on an interesting scientific topic which is rarely addressed in other existing simulation chambers. The manuscript aims to describe the chamber,

its equipment, its instrumentation, and to characterize the aerosol particle behavior (lifetimes), trace gas wall losses, and background levels of particles and trace gases. Furthermore, specific procedures for handling, aerosolizing, and sampling of bacteria are described and results of first test experiments on the viability of bacterial aerosol in the chamber are presented. Due to the focus on chamber characterization and first tests the scientific results are only of limited significance. Rather poor are the overall presentation quality, the English, and the scientific quality. While standard methods are described in great detail the applications or connection to the new simulation chamber are not given in a sufficient manner. Furthermore, several aspects of the tests and experiments are described insufficiently. Hence this manuscript should only be accepted for publication after major improvements.

Specific comments

Page 1 line 1: It isn't evident why and how the new simulation chamber facility can potentially contribute to aerosol modelling. Therefore the title is misleading. Either change the title or add a detailed explanation of the potential aerosol modelling link to the chamber.

Page 2 line 43: Explain specific what you mean. How can bacteria be chemically active in the atmosphere?

Page 2 line 44: Explain what you mean. How can bacteria favor the formation of condensation nuclei?

Page 2 line 54: Specify how many or which types of organisms can survive and what you mean with a long airborne transport. Give typical atmospheric transport or lifetimes.

Page 2 line 63-64: Explain which biogeochemical issue you mean.

Page 2 line 64-65: Explain what you mean with complex ecosystem.

Page 3 line 73: You should skip "mainly" in this sentence.

Page 3 line 74-75: The cited work is related to aerosol-cloud interaction but not to cloud chemistry.

Page 3 line 79-80: Be more precise what you mean.

Page 3 line 84-86: You should be more specific about the subjects of previous studies.

Page 3 line 93-96: Do you mean that ASCs with realistic simulation capabilities should be combined with biological facilities for adequate handling and characterization of bio aerosols? You should reformulate this sentence.

Page 3 line 104: Mention the modelling tools in this section.

Page 3 line 108: In Figure 1 the central ring has a height of 60 cm. Please be consistent.

Page 3 line 110: Figure 1 shows 4 flanges of 10 cm and 2 of 40 cm diameter. Please be consistent between text and figures.

Page 4 line 115: What do you mean with an ad-hoc metallic structure?

Page 4 line 121: Explain carefully if there are other means to retain pump oil to diffuse into the chamber. Explain why a two-step process to refill the chamber is needed.

Page 4 line 129: Give the fan speed in terms of revolutions per minute.

Page 4 line 131: Give the manufacturers of all components you mention in this section.

Page 4 line 138-139: Give positions and sensitivities for these sensors. How do these sensors interact with reactive trace gases like ozone?

Page 4 line 142: Are both lamps permanently installed or can the second one be installed on purpose? Give type and manufacturer for both lamps.

Page 5 line 160: It seems not necessary to me to describe an SMPS instrument in such detail.

Page 5 line 163: Reformulate this sentence.

Page 5 line 163: Note that a pre-impactor is required for a correct SMPS measurement to prevent false sizing due to multiple charged particles.

Page 5 line 170: I suppose the CPC is sensitive to particles larger than 4.5 nm.

Page 5 line 173: It is quite important to describe the design of the sampling lines and potential losses that could occur in them. E.g. sedimentation losses of larger aerosol particles in horizontal tubes.

Page 5 line 175: Explain how you calibrate your instruments and how you ensure their traceability.

Page 5 line 185-188: Explain how you distinguish between NO2 and NOx.

Page 5 line 190: I suppose you mean aerosol particle lifetime here. You must be precise with your language here since aerosol means a mixture of gas and particles.

Page 5 line 191: Some processes proceed on time scales of seconds.

Page 5 line 192: The manuscript describes the lifetime of NaCl particles within the simulation chamber for different particle sizes. Indeed an important characteristic for aerosol simulation chambers. However, the manuscript doesn't explain sufficiently how these lifetimes are defined, why the lifetimes for the different particle sizer are different, and the relevance of the lifetimes for experiments with typical bio aerosols. Several questions in this respect remain open. An important aspect is for example also the role of mixing in the chamber. How are the mixing times determined and how does the mixing fan influence the particle or trace gas lifetimes in the chamber? How is the mixing and particle lifetime influenced by injecting the sliding shelf? How broad is the particle lifetime distribution e.g. for bacterial aerosol particles ranging between 2.5 - 6.5 $\mu$m in length? The discussion of the possible time scales for studying typical bio aerosols in the new chamber compared to typical atmospheric residence or aging times is missing. Regarding Figure 3: You may combine the particle size measurements

done by the SMPS (mobility size) and OPC (optical size) instruments to obtain the geometrical particle size. Explain the very high particle lifetimes obtained only from analysis on the OPC data for the size range ∼300-500 nm. Are these data realistic? Explain how you calculate the uncertainties for size and lifetime and show them.

Page 6 line 197: Give type and manufacturer for the BLAM nebulizer. Consider adding this information to section 2.

Page 6 line 199: Explain what you mean with "a full range of particle dimension."

Page 6 line 202: Is the rotation speed of 5 Hz 5 revolutions per second? Explain how you determined the mixing time of 1 minute.

Page 6 line 204: How did you determine the mass decay curves?

Page 6 line 210: Replace aerosol lifetime by particle lifetime. According to figure 3 the NaCl particle lifetime ranges from about 1 h for particle diameters of 20 nm and 3 $\mu$m to about 10-15 h for particle diameters of 300 nm. Hence the 4 days are not justified. Which lifetimes did the two different bacteria strains have in your chamber? You should add this to figure 3 for comparison. What was the relative humidity for the lifetime studies with NaCl particles?

Page 6 line 214: No all trace gases are lost to the chamber walls.

Page 6 line 220: Typically, ozone wall losses also depend strongly on its concentration showing a bi-exponential behavior with much faster loss rates for the first few ppb.

Page 6 line 224: You should replace aerosol by particles here. How many particles (number & mass) are generated when you add ozone to the chamber and how does this change for subsequent experiments or after several cleaning cycles with high ozone concentrations.

Page 6 line 227: Reformulate this sentence better explaining the cleaning procedure.

Page 6 line 228: Reformulate this sentence and give the detection limit of your particle

measurements to define what no significant particle presence means. Didn't you count the particles directly with a CPC?

Page 6 line 228: Filling the simulation chamber with air from the laboratory through a HEPA filter can lead to changing amounts of trace gases in the chamber e.g. VOC which may impact the experiments and their reproducibility. An additional filter with an activated surface could improve this. Determination of the air quality in the chamber as well as controlling it constituents should be described in detail. Please note that it is not necessary to describe how a gas monitor works if you give type and maker but you should explain how you generate, dose and control the different gases including water vapour. The relative humidity is of special importance for many bioaerosols. Please note that the Humicap sensors typically suffer from exposure to higher ozone concentrations. Did you double check the humidity calibration after experiments with high ozone concentrations?

Page 7 line 233: Handling of bacteria is described in detail but it is not clear where this handling can be done and e.g. how quickly they can be transferred from the biological laboratory to the simulation chamber. It could be a unique strength of this simulation chamber facility e.g. if the handling would be possible in a nearby biological laboratory. Therefore, this aspect should be described in detail.

Page 7 line 246-247: Give a reference for this statement.

Page 7 line 270: How good could you estimate the number of cells.

Page 7 line 273: Explain OD600nm at first occurrence.

Page 8 line 278: Give the parameters in the equation and compare them to literature values.

Page 8 line 287: Give proper uncertainties for the CFU numbers and OD600nm values. Always use the same abbreviation throughout the text e.g. OD600nm not OD600. What was the OD600nm for E.coli.?

Page 8 line 289: Give the uncertainty for the "OD600 around 0.6" e.g. OD600nm of (0.6±0.3).

Page 8 line 290: What do you mean with excessive bacterial concentration?

Page 8 line 292-305: The technical details e.g. of the BLAM should be described in section 2 including the dimensions of the tubing.

Page 8 line 305: Define the nebulizing efficiency.

Page 8 line 307: Replace "tiny droplets" by a proper description of the droplet size distribution.

Page 8 line 309-310: Give the size distribution (e.g. mean diameter and standard deviation) generated and define what you consider respirable range. Explain why the respirable range is of interest here.

Page 9 line 321-329: How do you avoid contamination of the Petri dishes, as they seem to be exposed to laboratory air?

Page 9 line 335: Explain the gravitational settling method. What is the settling time distribution for the bacteria you studied? Compare the average settling times with the typical atmospheric residence times of those bacteria.

Page 9 line 336: Define in which respect you consider this method as efficient.

Page 9 line 338: Describe how the filter samples are collected.

Page 9 line 340: What do you mean with "tendency to aggregation"? Do you refer to sampling artifacts or to coagulation?

Page 9 line 348: Explain what kind of filter unit you used.

Page 10 line 369: Give an uncertainty for the estimated living bacteria concentration in the chamber e.g.: $(10^5 \pm ??)$ CFU m$^{-3}$ = 0.1 CFU cm$^{-3}$. Compare the number concentration of living and dead bacteria. Compare the number and size (mass)

[Figure]

concentrations of the aerosol particles measured with the dead and living bacteria concentrations.

Page 10 line 373: What do you mean with "statistically compatible", significant?

Page 10 line 377: Do you mean no significant effect related to RH? Would you expect a significant effect related to the variations of relative humidity? How could this be related to the residence time or drying time of the bacteria?

Page 10 line 382: Explain how the relative humidity in the chamber was increased and what you mean with the environmental value.

Page 11 line 395: Indicate if the uncertainty in the slope of the correlation ($\pm 5$ %?) includes the uncertainties of the individual measurement values in the plot. Replace "about 5 %" and "around 10 %" with well-defined values.

Page 11 line 399: Explain how the use of the optical density measurements influences the uncertainty of the cell quantification.

Page 11 line 404-408: Double check these observations by comparison with the particle measurements in the chamber.

Page 11 line 410: The conclusions should be reformulated and extended including a comparison of typical atmospheric residence times for bacteria with those that can be achieved within ChAMBRe.

Table 1: Explain the range of relative humidities.

Tabel 3: For which times during the experiments did you determine the relative humidities and temperatures?

Tabel 4: Can you estimate the ratio of CFU vs. non-CFU deposited on the Petri dishes e.g. based on the bacteria concentrations, sedimentation rate and area of the Petri dishes?

Figure 1: Explain the different parts of the chamber in the caption. Is there any air guiding tube surrounding the fan? Is the fan efficiently mixing the upper and typically warmer part of the chamber? What is the typical vertical temperature gradient? Would it be possible to heat the chamber to 37°C?

Figure 3: Indicate RH and temperature for the experiment in the caption.

Figure 4: Include uncertainties for the individual data points. Compare the optical density measurement to the CFU data and/or literature data.

Figures 5&6: Indicate if the uncertainties for the individual data points are included in uncertainties of the slopes.

Technical corrections:

Page 1 line 16: . . .processes at realistic but controlled conditions.

Page 1 line 21: . . .is made of stainless steel. . .

Page 1 line 22: . . .10 to 2 hours.

Page 1 line 24: . . . have impact on several levels as: . . .

Page 1 line 25: . . ., and geochemistry.

Page 2 line 40: . . .and maintain their pathogenic potential, . . .

Page 2 line 41-42: check wording

Page 2 line 44: . . .chemical, and biological properties. . .

Page 2 line 61-63: Reformulate sentence without brackets.

Page 3 line 109: . . .height.

Page 4 line 114: . . .designed to move specific samples inside the chamber as described. . .

Page 4 line 120: . . .failure it automatically closes in less than one ms, . . .

Page 4 line 127: . . .with four metallic arms of 25 cm length each. . .

Page 4 line 131: . . .and an accuracy of $\pm10\%$ of its reading. . .

Page 4 line 133: The pressure transducer contains. . .

Page 5 line 185: . . .concentrations are monitored. . .

Page 7 line 236-238: Reformulate the sentence in correct English.

Page 7 line 255: Reformulate the sentence in correct English.

Page 7 line 257: . . .prior to the injection.

Page 7 line 268: In both cases,. . .

Page 8 line 278: The cultivable cell concentration. . .

Page 8 line 288: . . .was prepared for nebulization. . .

Page 8 line 290: . . .was diluted (. . .

Page 8 line 306: . . .with a cavity depth and a cone diameter of . . .

Page 8 line 313-314: . . .completely separate the cylinder, which can be connected to the main chamber or. . ..

Page 9 line 316: This volume can be evacuated through a by-pass to the. . .

Page 9 line 325: Valve V2 is closed and the volume inside the pipe is flushed with clean air from the chamber.

Page 9 line 332: . . .bacteria have been injected. . .

Page 9 line 33: After exposure to the chamber atmosphere, . . .

Page 9 line 345: . . ..were not done in this case as the study. . .

Page 10 line 345: . . ..were not done in this case as the study. . .

Page 10 line 371: . . ..contributions.

Page 10 line 374: . . .appears to be adequate. . .

Page 11 line 402: . . .second set of experiments providing the Gram-negative microorganisms a more. . .
* * *

---

## Referee Comment (RC2) · Anonymous Referee #2 · 24 Aug 2018

The manuscript provides a characterisation of the new ChAMBRe atmospheric simulation chamber at Genova. The importance of the quantification of chamber-specific influences and interferences is not routinely recognised and the authors are to be commended on their attempts to provide such a characterisation. This manuscript provides the first part of "a user manual" for such a chamber and a demonstration of its fitness for purpose. As such, it is highly suitable for publication in AMT.

The range of important questions that can be addressed in ChAMBRe is succinctly and concisely summarised in the introduction, which thereby highlights the need for such a facility.

[Figure]

Key characteristics: Size dependent particle lifetime; Preservation of bioaerosol viability through injection and extraction; Ability to explore bioaerosol viability under variable controllable and measurable changes in atmospheric composition; Repeatability under clean conditions demonstrated for 2 bacterial strains; one each of gram positive and gram negative groups, showing that it is possible to investigate viability under changing environmental conditions.

The chamber and instrument description section is sufficiently detailed to provide the reader with the information to evaluate the suitability of the facility. It appears that the facility is very appropriately designed and well-appointed.

The chamber characterisation is one of the main foci of the paper and has been carried out and described well - certainly adequately for a reader to evaluate the characteristics of the facility.

The protocol for preparation and execution of the bacterial experiments has been well-developed and is described appropriately. As a non-specialist, I am not qualified to comment on the appropriateness of the biological handling protocols or e.g. choice of bacteria; however, for the purposes of the manuscript, the details provided are sufficient to reproduce the protocol.

The description of the first experiments was informative and provided a good illustration of the suitability for purpose of the facility. The repeatability within Poisson statistical expectation was convincing evidence for this. The fact that an empirically-determined dilution was required for E. coli (and that CFU was extremely RH dependent) is interesting and these experiments demonstrated the great care that will need to be employed in future investigations using the facility.

Generally, I find the publication very suitable for publication in AMT. However, I think there is one very important omission that can be addressed with modest discussion. This relates to the closeness to ambient conditions that is achievable within the facility. A key challenge in investigating PBAP in general and airborne bacteria in particular is

their extremely low ambient abundance (from 0.01 to 1 cells/cc). Such low concentrations present difficulties with particle sizing and counting instrumentation relating to the counting statistics, often physically limited by the instrument sample volume. I believe it is important that the authors discuss such limitations and the implications of necessarily studying under higher than ambient concentrations within ChAMBRe (such as differences in amount of reactant per cell). In this regard, it should be noted that the facility description indicates a maximum particle number measurement threshold of 10ˆ7 /cc, but now lower detection limit and sensitivity. Indeed, the only mention of the instrument sensitivity for the lower limit is on line 229 where the value is still not provided. A general discussion of the accessible ranges of concentrations, the challenges with instrumentation and the comparison to ambient conditions should be provided. One final point on this subject - one might have expected that a key instrument for PBAP experiments would be one of the recent online fluorescence instruments (e.g. WIBS, UV-APS or more recent developments). Could the authors discuss whether these factor in their plans. It would appear to provide an ideal opportunity for comparison of online and offline techniques and perhaps address some of the concentration concerns I have raised above.

There are a few additional points the authors could address:

line 98 and elsewhere: Genova or Genoa? Consistency should be ensured throughout the document

line 108 and 118: I presume this should be 5 * 10ˆ{-2} mbar, not 510ˆ{-2} (presumably for AMT, SI units should be used - the editor can advise). All subsequent pressure values also suffer from this notation and it should be corrected.

line 121: I presume the 2 step procedure is to ensure the HEPA and zeolite efficiency is not challenged by operating from 5 * 10ˆ{-2} mbar to 5 mbar. Is this correct? If so, it would be useful to state it.

line 142 - 147: I presume the 253 nm lamp is to allow sterilisation without ozone generation. If so, it would be useful to state it for the reader's benefit.

line 174 - 179: It is interesting to note the dehumidification system in the OPC. This will be very useful to avoid droplet ingress into the instrument and prevent too much hygroscopic growth. It will also ensure a good chance that non-spherical "solid" particles do not assume spherical geometry by water uptake. What is the implication of this for OPC sizing of e.g. rod-shaped or other non-spherical bacteria or dust? Don't OPCs rely on refractive index and shape assumptions?

line 191: I'd suggest the upper limit to typical reaction times should be days rather than hours (e.g. the gas phase oxidation lifetime of SO2, OH oxidation lifetime of methane, ageing of organic aerosol and increase in O:C ratio etc...)

line 198: I think the reference should be to 4.2 not 4.3

line 205: It is stated that "Aerosol dilution due to the air flow through the two counters (in total: 1.6 L min-1) was taken into account and properly corrected". Does this mean that pressure is held constant and the same amount of clean scrubbed air is supplied to the chamber? This should be stated.

line 208 - 209: "nicely reproduced" - please provide a more scientific description - a goodness of fit metric ideally

Trivial: I believe the Pasteur reference was originally from Annales des sciences naturelles, Zoologie, 4th series (1861), followed by its publication in Annales de chimie et de physique, 3rd series, 64 (1862), not 1890

---

## Author Comment (AC1) · 21 Sep 2018

The manuscript provides a characterisation of the new ChAMBRe atmospheric simulation chamber at Genova. The importance of the quantification of chamber-specific influences and interferences is not routinely recognised and the authors are to be commended on their attempts to provide such a characterisation. This manuscript provides the first part of "a user manual" for such a chamber and a demonstration of its fitness for purpose. As such, it is highly suitable for publication in AMT. The range of important questions that can be addressed in ChAMBRe is succinctly and concisely summarised

in the introduction, which thereby highlights the need for such a facility. Key characteristics: Size dependent particle lifetime; Preservation of bioaerosol viability through injection and extraction; Ability to explore bioaerosol viability under variable controllable and measurable changes in atmospheric composition; Repeatability under clean conditions demonstrated for 2 bacterial strains; one each of gram positive and gram negative groups, showing that it is possible to investigate viability under changing environmental conditions. The chamber and instrument description section is sufficiently detailed to provide the reader with the information to evaluate the suitability of the facility. It appears that the facility is very appropriately designed and well-appointed. The chamber characterisation is one of the main foci of the paper and has been carried out and described well - certainly adequately for a reader to evaluate the characteristics of the facility. The protocol for preparation and execution of the bacterial experiments has been well developed and is described appropriately. As a non-specialist, I am not qualified to comment on the appropriateness of the biological handling protocols or e.g. choice of bacteria; however, for the purposes of the manuscript, the details provided are sufficient to reproduce the protocol. The description of the first experiments was informative and provided a good illustration of the suitability for purpose of the facility. The repeatability within Poisson statistical expectation was convincing evidence for this. The fact that an empirically-determined dilution was required for E. coli (and that CFU was extremely RH dependent) is interesting and these experiments demonstrated the great care that will need to be employed in future investigations using the facility. Generally, I find the publication very suitable for publication in AMT. However, I think there is one very important omission that can be addressed with modest discussion.This relates to the closeness to ambient conditions that is achievable within the facility A key challenge in investigating PBAP in general and airborne bacteria in particular is their extremely low ambient abundance (from 0.01 to 1 cells/cc). Such low concentrations present difficulties with particle sizing and counting instrumentation relating to the counting statistics, often physically limited by the instrument sample volume. I believe it is important that the authors discuss such limitations and the implications of

necessarily studying under higher than ambient concentrations within ChAMBRe (such as differences in amount of reactant per cell). In this regard, it should be noted that the facility description indicates a maximum particle number measurement threshold of 10ËĘ7 /cc, but now lower detection limit and sensitivity. Indeed, the only mention of the instrument sensitivity for the lower limit is on line 229 where the value is still not provided. A general discussion of the accessible ranges of concentrations, the challenges with instrumentation and the comparison to ambient conditions should be provided. One final point on this subject - one might have expected that a key instrument for PBAP experiments would be one of the recent online fluorescence instruments (e.g. WIBS, UV-APS or more recent developments). Could the authors discuss whether these factor in their plans. It would appear to provide an ideal opportunity for comparison of online and offline techniques and perhaps address some of the concentration concerns I have raised above.

Authors reply: We thank the Reviewer for the extremely useful suggestions/criticisms and comments. We have carefully taken into account each point to improve the quality of the paper. In the text, we have added some information on typical bacteria residence time and concentration in the atmosphere as well as about minimum detection levels and sensitivity of used instrumentation. During our experiments, the typical bacteria concentration in the chamber is about 10ˆ5 cells/mˆ3 –see lines 369 of the original manuscript- (since 10ˆ7 is the concentration before the aerosolization process). This value is actually very close to the values of 0.01-1 cells/cc reported in literature (and by the reviewer) of typical ambient abundance. As the reviewer correctly notes, recent florescence instruments can help a lot in our experimental activity. In fact, the installation of a WIBS unit is expected in the next 6 months.

line 98 and elsewhere: Genova or Genoa? Consistency should be ensured throughout the document.

The format will be consistent in the revised manuscript.

line 108 and 118: I presume this should be 5 * 10Ë{-2} mbar, not 510Ë{-2} (presumably for AMT, SI units should be used - the editor can advise). All subsequent pressure values also suffer from this notation and it should be corrected.

The corrected notation will be used in the revised manuscript.

line 121: I presume the 2 step procedure is to ensure the HEPA and zeolite efficiency is not challenged by operating from 5 * 10Ë{-2} mbar to 5 mbar. Is this correct? If so, it would be useful to state it.

Actually, the two-step process comes from a long expertise with vacuum systems: in the very first phase of the venting, the use of nitrogen from a cylinder guarantees that no water vapor (or others) enter the chamber and penetrate in the walls. When the walls are "coated" with nitrogen, ambient air is used through the HEPA filter to bring back the chamber to atm. Pressure.

line 142 - 147: I presume the 253 nm lamp is to allow sterilisation without ozone generation. If so, it would be useful to state it for the reader's benefit.

Exactly, a clarification was added to the text.

line 174 - 179: It is interesting to note the dehumidification system in the OPC. This will be very useful to avoid droplet ingress into the instrument and prevent too much hygroscopic growth. It will also ensure a good chance that non-spherical "solid" particles do not assume spherical geometry by water uptake. What is the implication of this for OPC sizing of e.g. rod-shaped or other non-spherical bacteria or dust? Don't OPCs rely on refractive index and shape assumptions?

The "artifacts" in the OPC response related to particle shape are known and need proper correction (see for instance Caponi et al. (2017). Spectral- and size-resolved mass absorption efficiency of mineral dust aerosols in the shortwave spectrum: a simulation chamber study. ATMOSPHERIC CHEMISTRY AND PHYSICS, vol. 17, p. 7175) however, in our experiments we do not try to measure the bacteria size-distribution by

[Figure]

OPC (since the counts are totally dominated by the salt particles of the physiological solution) and the OPC is just used to check the global quantity of the injected material.

line 191: I'd suggest the upper limit to typical reaction times should be days rather than hours (e.g. the gas phase oxidation lifetime of SO2, OH oxidation lifetime of methane, ageing of organic aerosol and increase in O:C ratio etc...)

The statement will be corrected in the revised manuscript. .

line 198: I think the reference should be to 4.2 not 4.3

Done.

line 205: It is stated that "Aerosol dilution due to the air flow through the two counters (in total: 1.6 L min-1) was taken into account and properly corrected". Does this mean that pressure is held constant and the same amount of clean scrubbed air is supplied to the chamber? This should be stated.

The chamber is designed to ensure that the pressure is kept constant: the same amount of clean air is introduced into the chamber through the input from the HEPA filter.

This statement will be included in the revised manuscript.

line 208 - 209: "nicely reproduced" - please provide a more scientific description – a goodness of fit metric ideally.

The Lai & Nazarov model can of course predict the life-times in a approximative way (but we also must underline that what it is really important to know is the typical lifetime values for particles in a given dimensional range). As a matter of fact, "nicely" here corresponds to a mean discrepancy between calculated and measured lifetimes of about 50%. This figure will be included in the revised text changing the sentence in:

"…nicely reproduced (i.e. the mean discrepancy between measured and calculated values is around 50%)".

[Figure]

Trivial: I believe the Pasteur reference was originally from Annales des sciences naturelles, Zoologie, 4th series (1861), followed by its publication in Annales de chimie et de physique, 3rd series, 64 (1862), not 1890

Citation will be corrected in the revised manuscript.

---

## Author Comment (AC2) · 21 Sep 2018

Anonymous Referee #1 Review of the manuscript amt-2018-147 with title: "ChAMBRe: a new atmospheric simulation Chamber for Aerosol Modelling and Bio-aerosol Research" by Massabò et al.

[Figure]

General comments:

This manuscript describes a new atmospheric simulation chamber and its potential use to simulate the interaction of trace gases and biological aerosol particles like bacteria. Therefore, it fits well in the scope of the journal of atmospheric measurement techniques and focusses on an interesting scientific topic which is rarely addressed in other existing simulation chambers. The manuscript aims to describe the chamber, its equipment, its instrumentation, and to characterize the aerosol particle behavior (lifetimes), trace gas wall losses, and background levels of particles and trace gases. Furthermore, specific procedures for handling, aerosolizing, and sampling of bacteria are described and results of first test experiments on the viability of bacterial aerosol in the chamber are presented. Due to the focus on chamber characterization and first tests the scientific results are only of limited significance. Rather poor are the overall presentation quality, the English, and the scientific quality. While standard methods are described in great detail the applications or connection to the new simulation chamber are not given in a sufficient manner. Furthermore, several aspects of the tests and experiments are described insufficiently. Hence this manuscript should only be accepted for publication after major improvements.

We thank the Reviewer for the extremely useful suggestions/criticisms and comments. We have carefully taken into account each point to improve the quality of the paper.

Specific comments

Page 1 line 1: It isn't evident why and how the new simulation chamber facility can potentially contribute to aerosol modelling. Therefore the title is misleading. Either change the title or add a detailed explanation of the potential aerosol modelling link to the chamber.

The Referee is right. Actually, the title contains the explanation of the ChAMBRe acronym and we fear it could be difficult to change. We can however include the following explanation at the end of sec. 1.2:
While ChAMBRe, as other ASCs, is a multi-purpose facility, the outcomes of the correlation between bacteria viability and atmospheric condition/composition will provide the input for developing ad-hoc modules to be then implemented in chemical transport models. This can be done following a scheme often used for the chemical mechanisms parameterization (see for example the smog chamber experiments used for the evaluation of Carbon Bond mechanisms in Parikh H. M. et al., "Evaluation of aromatic oxidation reactions in seven chemical mechanisms with an outdoor chamber" Environ. Chem. 2013, 10, 245–259). Such software tools, are widely used both in scientific research and in air quality evaluations, to predict the fate (i.e. transport, deposition and chemical changes) of the atmospheric pollutants and, at the moment, they do not include any biological patch.

Page 2 line 43: Explain specific what you mean. How can bacteria be chemically active in the atmosphere?

It has been proposed that bio-aerosols have a potential role in the chemistry of organic compounds in the troposphere via microbiological degradation and hence inducing changes in the IN or CCN ability of organics in atmosphere (Ariya and Amyot, 2004).

This statement will be included in the revised manuscript in sec. 1.1

Page 2 line 44: Explain what you mean. How can bacteria favor the formation of condensation nuclei?

Bauer et al. (2003) suggested that the chemical composition, structure and hydrophilicity of the surface layer of bacteria could play important roles in CCN activity.

This statement will be included in the revised manuscript in sec. 1.1.

Page 2 line 54: Specify how many or which types of organisms can survive and what you mean with a long airborne transport. Give typical atmospheric transport or lifetimes.

Airborne bacterial communities are highly diverse, and variations in their species diversity are quite complex. The bacterial composition in air is strongly dependent on many factors such as seasonality, meteorological factors, anthropogenic influence, variability of bacterial sources and many other variables. Still, the general trend from the literature is that bacteria found in the air often belong to groups that are also common in the soil (e.g. Firmicutes, Proteobacteria, Actinobacteria) (Després et al., 2012). Due to their small size, bacteria have a relatively long atmospheric residence time (on the order of several days or more) compared to larger particles and can be transported over long distances (up to thousands of kilometers). Measurements show that mean concentrations in ambient air can be greater than 1x10ˆ4 cells mˆ-3 over land, whereas concentrations over the sea may be lower by a factor of 100-1000 (Burrows et al., 2009a, Burrows et al., 2009b).

This statement will be included in the revised manuscript in sec. 1.1.

Page 2 line 63-64: Explain which biogeochemical issue you mean.

We simply mean the fact, reported and described in the quoted literature and in other works, that particulate matter transported from desert areas to the parts of the world, is made not only by dust but it contains several biological "particles" and bacteria in particular. As a matter of fact, bacteria can stick the dust particles and can be more efficiently (i.e. remaining viable) transported through long distances. We just quoted the references in the text but we can easily add a few words to make the point more clear.

Page 2 line 64-65: Explain what you mean with complex ecosystem.

Actually, we tried to wrap-up the previous points just underlying the fact that bacteria are influenced by the atmospheric conditions but they also contribute to modify the atmosphere. An ecosystem is a biological community of interacting organisms and their physical environment, such as air, water, and mineral soil. Ecosystems include interactions among organisms, and between organisms and their environment, through very complex mechanisms. However, we agree that the statement is too generic and

we propose to delete it. The information of sec. 1.1 does not change.

Page 3 line 73: You should skip "mainly" in this sentence.

This will be done in the revised manuscript.

Page 3 line 74-75: The cited work is related to aerosol-cloud interaction but not to cloud chemistry.

We propose to modify the text as follows:

ASCs have been used to study chemical and photochemical processes that occur in the atmosphere, such as ozone formation (Carter et al., 2005 and references therein) and cloud chemistry (Wagner et al., 2006) or aerosol-cloud interaction (Benz et al., 2005), etc.

Page 3 line 79-80: Be more precise what you mean.

We propose to modify the text as follows:

Since the interplay of bio-aerosol and atmospheric conditions is still poorly known, suitable facilities are needed, where transdisciplinary studies gathering atmospheric physics-chemistry and biology issues are possible.

Page 3 line 84-86: You should be more specific about the subjects of previous studies.

We have reported these studies just to cite the principal topic related to atmospheric chambers i.e. ice nucleation and cloud condensation.

We propose to modify the sentence in the revised manuscript as follows:

The use of atmospheric simulation chambers has been much more limited and focussed on the interaction of bacteria with atmospheric parameters, regarding bio-aerosols release effects (Jones and Harrison, 2004), and on ice nucleation and cloud condensation (Möhler et al., 2008; Bundke et al., 2010; Chou, 2011).

Page 3 line 93-96: Do you mean that ASCs with realistic simulation capabilities should

**AMTD**

[Figure]

be combined with biological facilities for adequate handling and characterization of bio aerosols? You should reformulate this sentence.

We agree that our manuscript was not clear enough on this point. We could change the statement in this way:

Such experimental evidence made clear that the effects of atmospheric pollution on bacteria viability could be studied in atmospheric chambers. In order to perform systematic studies to resolve and describe the physical and chemical mechanisms ruling these interactions, dedicated facilities with a microbiology laboratory linked to the ASC for the handling and characterization of bio-aerosol are needed.

Page 3 line 104: Mention the modelling tools in this section.

Actually, the ChAMBRe task in EUROCHAMP is double-fold: the development of a protocol to perform experiments in ASCs with bacteria and, more in general, bio aerosol and the collection of data (through a set of experiments) to correlate bacteria viability and atmospheric conditions. The latter should results, in two year from now, in the assessment and implementation of specific routines/patches to be inserted in chemical transport models (that should/could so evolve to BCTM). This is a very ambitious program, likely longer than the EUROCHAMP time frame.

We agree that the statement in the manuscript is too sharp and, to not be too long and to remain in the aims of the present work, we could modify the statement just in:

...is one of the nodes of the EUROCHAMP-2020 network with specific tasks on bio-aerosol studies.

Page 3 line 108: In Figure 1 the central ring has a height of 60 cm. Please be consistent.

The right number is 60, the text will be changed accordingly.

Page 3 line 110: Figure 1 shows 4 flanges of 10 cm and 2 of 40 cm diameter. Please

be consistent between text and figures.

Numbers in the text were wrong and will be corrected (2 x 40 and 4 x 10).

Page 4 line 115: What do you mean with an ad-hoc metallic structure?

The lower dome is hold by a metallic support to maintain the entire structure in vertical position.

This statement will be included in the revised manuscript.

Page 4 line 121: Explain carefully if there are other means to retain pump oil to diffuse into the chamber. Explain why a two-step process to refill the chamber is needed.

The statement refers to the "safety" equipment. In the normal operation, the chamber is also equipped with a manual valve (positioned between the chamber and the gate valve). Before quitting the pumps such manual valve is closed so ensuring that no oil can diffuse in the chamber during the slowing-down phase. We do not considered to add this detail in the text but we can of course add a proper statement.

The two-step process comes from a long expertise with vacuum systems: in the very first phase of the venting, the use of nitrogen from a cylinder guarantees (or anyway help) that no water vapor (or others) possibly entering the chamber can penetrate in the walls. When the walls are "coated" with nitrogen, ambient air is used through the HEPA filter to bring back the chamber to atm. pressure.

Page 4 line 129: Give the fan speed in terms of revolutions per minute.

This will be done in the revised text.

Page 4 line 131: Give the manufacturers of all components you mention in this section.

This piece of information will be included in the revised text.

Page 4 line 138-139: Give positions and sensitivities for these sensors.

This piece of information will be included in the revised text.
How do these sensors interact with reactive trace gases like ozone?

The models selected for ChAMBRe are, according to the data sheets, resistant to reactive gases.

Page 4 line 142: Are both lamps permanently installed or can the second one be installed on purpose?

The second one can be installed when needed.

Give type and manufacturer for both lamps.

This piece of information will be add at the revised manuscript.

Page 5 line 160: It seems not necessary to me to describe an SMPS instrument in such detail.

We could shorten a little bit the length in the revised manuscript, however we believe that the given information could help the reader not familiar with such instrument.

Page 5 line 163: Reformulate this sentence.

The sentence could be reshaped in this way.

The DMA is available with two different columns, working alternatively in the size range 5.5-350.4 nm (MDMA), and 11.1-1083.3 nm (LDMA), and classifying particles in 50 dimensional classes.

Page 5 line 163: Note that a pre-impactor is required for a correct SMPS measurement to prevent false sizing due to multiple charged particles.

The pre-impactor is routinely used, we did not include this detail in the text but we can better specify if needed.

Page 5 line 170: I suppose the CPC is sensitive to particles larger than 4.5 nm.

Yes of course, the right wording is "larger than 4.5 nm"

Page 5 line 173: It is quite important to describe the design of the sampling lines and potential losses that could occur in them. E.g. sedimentation losses of larger aerosol particles in horizontal tubes.

The SMPS has been connected to ChAMBRe through a smoothly bended pipe in a way to have a horizontal length of about 10 cm followed by a vertical part of about 30 cm. The OPC, which counts larger particles is connected to ChAMBRe by an ad-hoc set-up with the inlet directly sucking from one of the large flanges: no horizontal tubes (actually no tubes at all).

These detail will be added to the revised text.

Page 5 line 175: Explain how you calibrate your instruments and how you ensure their traceability.

The OPC is sent back to the factory for re-calibration at regular period: it has been calibrated just before the experiments described in the manuscripts.

Page 5 line 185-188: Explain how you distinguish between NO2 and NOx.

The chemiluminescence principle is used for automatic monitoring of NO, NOx and NO2 in ambient air. The reaction between NO and O3, which is the basis for the CLD (chemiluminescence detector), emits photons that are detected by a cooled photomultiplier tube (PMT).

$NO + O3 \rightarrow NO2^* + O2 \quad NO2^* \rightarrow NO2 + h\upsilon$

The CLD output voltage is proportional to the NO concentration. In order to measure the NO2 concentration, the sample must first be reduced into NO, and this is achieved with a heated molybdenum NOx converter:

$Mo + 3NO2 \rightarrow MoO3 + 3NO$

NO cycle: The sample moves directly into the reaction chamber (of the instrument) where NO oxidation by ozone takes place. The photomultiplier tube signal, minus the

black signal, is proportional to NO molecule number within the sample. For the NOx cycle the sample passes through the converter oven which reduces NO2 to NO, then it is mixed with ozone in the reaction chamber. The photomultiplier tube signal, minus the black signal, is proportional to the sum of NO and NO2 molecule (reduced to NO in the converter) contained in the sample.

A shorter explanation will be included in the revised manuscript.

Page 5 line 190: I suppose you mean aerosol particle lifetime here. You must be precise with your language here since aerosol means a mixture of gas and particles.

Yes, we'll use "particle aerosol lifetime" as section 3.1 title.

Page 5 line 191: Some processes proceed on time scales of seconds.

The mistake will be corrected in the revised manuscript

Page 5 line 192: The manuscript describes the lifetime of NaCl particles within the simulation chamber for different particle sizes. Indeed an important characteristic for aerosol simulation chambers. However, the manuscript doesn't explain sufficiently how these lifetimes are defined, why the lifetimes for the different particle sizer are different, and the relevance of the lifetimes for experiments with typical bio aerosols. Several questions in this respect remain open. An important aspect is for example also the role of mixing in the chamber. How are the mixing times determined and how does the mixing fan influence the particle or trace gas lifetimes in the chamber? How is the mixing and particle lifetime influenced by injecting the sliding shelf? How broad is the particle lifetime distribution e.g. for bacterial aerosol particles ranging between 2.5 - 6.5 $\mu$m in length? The discussion of the possible time scales for studying typical bio aerosols in the new chamber compared to typical atmospheric residence or aging times is missing. Regarding Figure 3: You may combine the particle size measurements done by the SMPS (mobility size) and OPC (optical size) instruments to obtain the geometrical particle size. Explain the very high particle lifetimes obtained only from

analysis on the OPC data for the size range _300-500 nm. Are these data realistic? Explain how you calculate the uncertainties for size and lifetime and show them.

We try to answer point-to-point:

"How lifetimes are defined?"

Particles concentration in the chamber decreases with time due to effects: dilution and wall deposition. With a sucking flow rate (i.e. the sum of the working flows of all the equipment connected to the chamber) constant in time (ïĄȨ), the particle concentration trend can be described by a simple equation: C(t) = C(0)e-kt , with k = ïĄ́c + ïĄȨ/V (V = chamber volume). The term ïĄ́cïĂÃ̆which summarizes the wall losses effects, is the inverse of the particle life-time. We did not include this definition in the text since we considered this point as been clarified by previous literature studies but we can easily add a few lines in the revised manuscript. Moreover, the procedure to measure "life-times" have been described in previous literature paper and it has also been assessed within the EUROCHAMP2020 consortium, basically as shown in Fig. 1.

"why the lifetimes for the different particle sizer are different ?"

We do not have a completely firm explanation. We want emphasize that the uncertainties given in Figure 3 contain the statistical part only. The region where the two sizers overlap is quite narrow and the discrepancy is concentrated between 300 and 600 nm i.e. the particle size corresponding to the longer life-times we could appreciate. In such interval, the statistics in the SMPS data is very low, as can be inferred by the large error bars plotted in figure 3. We cannot completely exclude that other contributes to the error budget could come, for instance, from background subtraction/fluctuation. On the contrary, the OPC counts are pretty high and no significant statistics uncertainty affects those results (but the error bars are plotted. . .see also below our answer to your comment on Fig. 3). However, even if in this case the geometric diameter differ from less than 10% from the optical one, it is known that the first bins of the Grimm OPC could suffer of systematic effects (Santi et al., 2010. Real-time aerosol photometer and optical particle counter comparison. Nuovo Cimento B 125(8):969 -981). Putting all these things together, we must actually admit that the results in the named size range could be more uncertain than as shown in the plot and that the manuscript did not make this point clear enough. We did not further investigate this issue since we were focused in assessing the typical values of "bacteria lifetime" (particles > 2 microns) in ChAMBRe.

We propose to modify the text in the manuscript as follow:

Particles lifetime in ChAMBRe varies from few hours to about 1 day depending on particle size. The uncertainty on particles life-time plotted in Figure 3 has been evaluated on a pure statistical basis. Actually, in the size region between 300 and 600 nm, both the SMPS and OPC data could be particularly sensitive to other effects (e.g. background fluctuation for the SMPS, systematic artifacts in the first OPC bins) which have not been fully investigated in this work and that do not change the typical feature depicted in Figure 3.

The caption of Figure 3 will be updated adding the sentence: Error bars include statistical uncertainties only. "what's the relevance of the lifetimes for experiments with typical bio aerosols ?"

while the life-time global picture is part of a more general characterization of the chamber performance, we consider the values in the range 1 – 3 ïA■m as indicative of the possible bacteria life-time. We mean that bacteria are subjected not only to wall deposition but the number of the viable units would decrease with time for other reasons. So, the life-time of viable bacteria cannot be simply inferred from the data in Figure 3 which could, in a certain way, be seen as "upper limit" of the bacteria life-time and hence of the effective time interval to run any experiment. Actually, as reported in sec.4 we decided, for the very first tests reported in the manuscript, to limit the exposure time of the petri dishes to a maximum of 5 hours i.e. a life-time of particles slightly smaller than 1 ïA■m.

We could add a proper sentence in the text even if we would prefer to keep separate the

discussion on the general performance/characterization from the issues directly linked to bacteria.

How are the mixing times determined and how does the mixing fan influence the particle or trace gas lifetimes in the chamber?

See in the following our answer to the comment at page 6, line 202.

How is the mixing and particle lifetime influenced by injecting the sliding shelf?

Aerosol particle life-time reported in fig. 3 have been measured without inserting the sliding shelf. This way, they represents the general feature of the chamber. When the shelf is inserted to collect bacteria, it certainly produces an effect (likely a life-time reduction) that we are going to assess through a complete fluid dynamic calculation (in progress but it will take a few months). However, when we use the shelf we must be sure to maintain it in the chamber for a time long enough to collect all the viable bacteria and the 5-hour upper limit quote above goes in this direction.

"How broad is the particle lifetime distribution e.g. for bacterial aerosol particles ranging between 2.5 - 6.5 $\mu$m in length?"

According to the data reported in Fig. 3, the life-time of particles in the quoted range varies (roughly) from 1 and 3 hours. So far, we cannot collect any direct information on bacteria lifetime (we are working in this direction introducing further time-sensitive collection methods) and, again, this is part of the arguments that brought us to select a 5-hour exposure time for the Petri dishes during the experiments with bacteria. We have to add that, after each experiment, a new set of Petri had been inserted in the chamber looking for residual bacteria but we never observed any sizeable signal. So, (at the moment) we can conclude that the life-time of viable bacteria (both the strains) is lower than 5 hour.

The discussion of the possible time scales for studying typical bio aerosols in the new chamber compared to typical atmospheric residence or aging times is missing.

We have considered this point above (Referee comment on line 54, pag 2). We are aware that the life-time in the chamber is much shorter than the typical (or possible) residence time in the atmosphere however our aim is to have time enough to study the impact on the bacteria viability of specific pollution levels. This seems to be a realistic goal considering the results of the pilot experiment performed at CESAM (where life-times are very similar to those detected in ChAMBRe) by Brotto et al., 2015. The program of systematic experiment we are going to start will assess this point, furthermore the fluid dynamic calculation that we are performing (actually, some Colleagues from the Engineering Dept.) we'll hopefully highlight solutions to increase the life-time.

Regarding Figure 3: You may combine the particle size measurements done by the SMPS (mobility size) and OPC (optical size) instruments to obtain the geometrical particle size. Explain the very high particle lifetimes obtained only from analysis on the OPC data for the size range _300-500 nm. Are these data realistic? Explain how you calculate the uncertainties for size and lifetime and show them.

Uncertainties are calculated during the fitting (weighted on the statistic uncertainty of each value) of the exponential decay trend of particle concentration measured by SMPS/OPC. They are correctly reported in Fig. 3 and in some cases, the error bar length is comparable with the dimension of the points in the plot (and therefore are difficult to see but they can appreciated when zooming Fig. 3). On the horizontal axis, points are positioned at the center of each size bin. The OPC data in the range 300-500 nm have very low statistical uncertainty (at least when compared with the corresponding SMPS values) and therefore much smaller error bars. However, it is known that the first channels of the Grimm-OPC can suffer of systematics uncertainty which could affect the results. We have to say that a similar study performed at CESAM resulted in life-time in the quoted interval of about 3 days (in that case OPC data were available from 400-500 nm and, more in general, with number very similar to those here reported, see https://www.eurochamp.org/Facilities/SimulationChambers/CESAM.aspx). We agree that this part of the manuscript is too sharp and that in the revised manuscript

we should comment that the OPC data at diameters lower than 500 nm are not completely firm and that, conservatively, we consider maximum life-time in ChAMBRe of about 1 day for particles around 300 nm.

Page 6 line 197: Give type and manufacturer for the BLAM nebulizer.

This piece of information will be added in the revised manuscript.

Consider adding this information to section 2.

We could add this information in sec. 2 even if we'd prefer to maintain separate the discussion on bio-aerosol equipment.

Page 6 line 199: Explain what you mean with "a full range of particle dimension."

We agree the statement is too generic. The BLAM can generate poly-disperse aerosol (or it can be used to nebulize particles dispersed in a liquid solution) up to the micrometric range. We could actually detected injected particles and/or bacteria up to 6 ïA■m.

We could modify the sentence in the text in "up to the micrometric range"

Page 6 line 202: Is the rotation speed of 5 Hz 5 revolutions per second? Explain how you determined the mixing time of 1 minute.

Yes, 5 Hz = 5 rev/sec = 300 rpm. We'll modify the units in r.p.m. in the revised manuscript.

We performed several experiments to determine gas phase mixing time at different fan speed. To do so, we used nitrogen monoxide as a tracer. Firstly, we checked that nitrogen monoxide is sufficiently inert inside our chamber, to perform this kind of studies. It was injected into the chamber at atmospheric pressure and 25 °C and monitored by using the AC32e monitor installed at ChAMBRe with a sampling flow rate of 0.66 L min-1. The inlet line was mounted on a lateral port in the upper dome of chamber. The Fig. 2 shows that the tracer injected in the chamber can be considered as well-

mixed in less than 60 s at the maximum of fan speed (600 rpm). This mixing time is relatively short comparing to the experiment durations (which may last for several hours). Nevertheless, when designing experiments, one will have to take into account this information. In the manuscript we did a material mistake quoting at 5Hz=300 rpm the mixing time actually measured at 10 Hz = 600 rpm. We'll change the sentence in the revised manuscript as "in a mixing time of about 2 minute".

Page 6 line 204: How did you determine the mass decay curves?

See above our answer to Referee question on line 192, page 5.

Page 6 line 210: Replace aerosol lifetime by particle lifetime.

This correction will be done in the revised manuscript.

According to figure 3 the NaCl particle lifetime ranges from about 1 h for particle diameters of 20 nm and 3 $\mu$m to about 10-15 h for particle diameters of 300 nm. Hence the 4 days are not justified.

This is true and the "4-day" is due to a material mistake. The revised manuscript will report a consistent lifetime value with what is shown in figure 3 (i.e. max. around 1 day).

Which lifetimes did the two different bacteria strains have in your chamber? You should add this to figure 3 for comparison.

As a result of a preliminary test of bacteria time-segregated collection, bacteria life time in the chamber is expected to be lower than 5 hours, according with the particles lifetime plot. At the moment we do not have any on-line monitor of bacteria concentration (a WIBS unit is on the shopping list and a budget request has been submitted including this item). We inject (sec. 4) bacteria in a NaCl solution, hence during the experiments reported in the manuscript the particle sizers signal was completely dominated by salt particles. As reported above, we controlled that after a 5-hour exposure, all the viable bacteria get collected on the Petri dishes but at the moment we are not in the condition

to answer precisely to the Referee's question.

What was the relative humidity for the lifetime studies with NaCl particles?

It was about 47%, we'll include this piece of information in the revised manuscript.

Page 6 line 214: No all trace gases are lost to the chamber walls.

We'll correct the sentence in "as the gaseous species can be lost to the chamber walls"

Page 6 line 220: Typically, ozone wall losses also depend strongly on its concentration showing a bi-exponential behavior with much faster loss rates for the first few ppb.

We fitted the O3 concentration decay curves cutting-off the first part of the trend (i.e. the first 15 min) to avoid possible artefacts and actually we did not observe a double-exponential decay, at least in the range (of initial concentration) 300-1000 ppb quoted in the manuscript. We could maybe see a faster decrease in the first minutes = first 10 -15 ppb but we have basically neglected this part.

Page 6 line 224: You should replace aerosol by particles here.

This will be corrected in the revised manuscript.

How many particles (number & mass) are generated when you add ozone to the chamber and how does this change for subsequent experiments or after several cleaning cycles with high ozone concentrations.

As we wrote in the manuscript, we used ozone just to sterilize the chamber after each experiment with bacteria and in between two vacuum cycles. Actually, we monitored the particle background level after the cleaning procedure and at the beginning of each experiment and till now it remained in the range indicated below (see answer to comment at Page 6 line 228). By the way, we cannot observe any particle formation in the OPC range while producing O3 (so far, we preferred to have the SMPS not working during the cleaning procedure with O3).

Page 6 line 227: Reformulate this sentence better explaining the cleaning procedure.

We'll reformulate the sentence in this way:

After each experiment, the chamber is cleaned by a multi-step procedure: the UV lamp (see sec. 2.1) is first switched on for10 min, the chamber is then evacuated and vented to atmospheric pressure through an HEPA filter (section 2.1). Afterwards, a high ozone concentration (>500 ppb) is produced to be sure to sterilize any part of the set-up possibly not reached before by the UV rays. Finally, the chamber is evacuated and vented again.

Page 6 line 228: Reformulate this sentence and give the detection limit of your particle measurements to define what no significant particle presence means. Didn0t you count the particles directly with a CPC?

The statement will be reformulated as follow:

Background level measurements performed subsequently to chamber cleaning showed no significant particles presence (i.e. about 2 and 0.5 particle cmˆ-3, respectively in the SMPS-LDMA and OPC range).

Page 6 line 228: Filling the simulation chamber with air from the laboratory through a HEPA filter can lead to changing amounts of trace gases in the chamber e.g. VOC which may impact the experiments and their reproducibility. An additional filter with an activated surface could improve this. Determination of the air quality in the chamber as well as controlling it constituents should be described in detail. Please note that it is not necessary to describe how a gas monitor works if you give type and maker but you should explain how you generate, dose and control the different gases including water vapour. The relative humidity is of special importance for many bioaerosols. Please note that the Humicap sensors typically suffer from exposure to higher ozone concentrations. Did you double check the humidity calibration after experiments with high ozone concentrations?

[Figure]

Actually, as reported at the end of sec. 2.1, a zeolite trap is mounted upstream the HEPA filter. We do not have at the moment a VOC monitor even if we are aware of its importance (as the WIBS, it is on the shopping list for next future). This, in our opinion, does not impact on the first results reported in the manuscript in sec. 4, where we simply show that the procedure to grow, inject and collect viable bacteria is well under control. It's clear however that VOC as well other parameters will be important for the next phases of our program. The Humicap sensor has been selected (see comment above in answer to your comment on Page 4 line 138-139). We checked the Humicap by comparing its output with another unit (used in the lab surrounding ChAMBRe) never exposed to ozone.

Page 7 line 233: Handling of bacteria is described in detail but it is not clear where this handling can be done and e.g. how quickly they can be transferred from the biological laboratory to the simulation chamber. It could be a unique strength of this simulation chamber facility e.g. if the handling would be possible in a nearby biological laboratory. Therefore, this aspect should be described in detail.

We propose to add at the beginning of sec. 4.2 the following text: Several techniques for bacteria and bio-aerosol characterization are available on site. In the same building that hosts the atmospheric simulation chamber there is a basic microbiology lab equipment allowing for culture analysis in vitro (isolation, identification, growth) and biochemical tests (e.g. catalase and oxidase): autoclave (Asal mod.760), vortex, centrifuge and micro-centrifuge (Eppendorf centrifuge 5417R), water purification system Milli-Q (Millipore-Elix), incubator for temperature control Ecocell and Friocell MMM Group, Steril-VBH Compact "microbiological safety" cabinet, Thermo electron corporation steri-cycle HEPA Class 100 incubator; optical microscope (Nikon Eclipse TE300) for bacterial detection and live/dead discrimination by epifluorescence with specific dyes and for immunoassay fluorescence to label antigenic bacterial target, fluorescent molecule or enzyme. The transfer of bacteria from the biological laboratory to the simulation chamber takes only a few minutes, ensuring a quickly execution of the

chamber experiments, once the desired phase of bacteria growth is reached, and then a quick treatment of the samples collected after the experiments in the chamber.

Page 7 line 246-247: Give a reference for this statement.

We'll add a reference to Earl et al., 2008

Page 7 line 270: How good could you estimate the number of cells.

Data, obtained from spectrophotometric measurements (OD600 nm), were used to estimate when the mid-exponential phase (corresponding an OD600nm of 0.5) is reached, not to determine the cells concentration. The number of cultivable cells is determinate as Colony Forming Units (CFU), by standard dilution plating. The uncertainties for the CFU numbers are reported in tables. We understand however that the text in the manuscript was not clear enough and we propose to modify the text as follow:

The optical density of the bacterial solution, measured at a wavelength of 600 nm, is a common method for estimating the concentration of bacterial cells in a liquid. The amount of the light scattered by the microorganisms suspension is an indication of the biomass contents (Sutton, S. 2011). Data, obtained from spectrophotometric measurements (OD600nm), were used to estimate when the mid-exponential phase (corresponding an OD600nm of 0.5) is reached. Actually, the number of cultivable cells was counted as Colony Forming Units (CFU), by standard dilution plating etc. etc.

Page 7 line 273: Explain OD600nm at first occurrence.

See the answer to the previous comment.

Page 8 line 278: Give the parameters in the equation.

The parameters are: (B. subtilis curve, a is 1.10 ± 0.01, b is 38 ± 2; E. coli curve, a is 0.83 ± 0.01 and b is 41 ± 1). This information will be added in the revised manuscript.

and compare them to literature values.

We did not find any literature value to be compared with these numbers. Actually, according to the opinion of the microbiologists in the group, this is a piece of information usually neglected.

Page 8 line 287: Give proper uncertainties for the CFU numbers and OD600nm values.

Always use the same abbreviation throughout the text e.g. OD600nm not OD600.

The abbreviation will be harmonized in the revised text. Uncertainty on CFU/ml in each single experiment are fully reported in Tables 2 and 4. The sentence here aims simply to clarify the order of magnitude of the working conditions

What was the OD600nm for E.coli.?

Values and uncertainties of OD600nm for E. coli is given in Table 4. The text will be modified as follow:

In particular, for E. coli, to obtain the final concentration of 106 CFU mL-1, the initial cells suspension with an OD600nm around 0.6 (single values are reported in Table 4) was diluted etc. etc.

Page 8 line 289: Give the uncertainty for the "OD600 around 0.6" e.g. OD600nm of (0.6±0.3).

The expression "OD600nm around 0.6" indicates that in each experiment we tried to reach the same OD value, to obtain bacteria cells indicatively at the same growth phase. The OD values for each experiment are shown in Table 4 where we'll add, in the revised manuscript, the corresponding uncertainties. Typical values for uncertainties are 0.03.

Page 8 line 290: What do you mean with excessive bacterial concentration?

We actually try to avoid an excessive bacterial concentration of colonies counted on each petri dishes as explained in the paragraph 5.2. We propose to reformulate the sentence as follow:

to avoid an excessive bacterial concentration on the Petri dishes exposed inside the Chamber (see the paragraph 5.2).

Page 8 line 292-305: The technical details e.g. of the BLAM should be described in section 2 including the dimensions of the tubing.

As reported above we'd prefer to have the BLAM description in sec. 4.2 but we can easily move this part of the text to sec.2. In any case the sentence on the tubing will be modified as follow:

...connected to the chamber with a curved stainless-steel tube (length = 50 cm, diameter = 1.5 cm).

Page 8 line 305: Define the nebulizing efficiency.

The nebulization efficiency is defined as the ratio between the mass of the produced aerosol to the mass of the solute or of the material suspended in the liquid inserted in the BLAM.

We propose to modify the text as follow:

with a nebulization efficiency (i.e. mass ratio between the mass of the produced aerosol to the mass of the solute or of the material suspended in the liquid inserted in the BLAM) between 1 % and 8 %.

Page 8 line 307: Replace "tiny droplets" by a proper description of the droplet size distribution.

We give this information in the answer to the next comment.

Page 8 line 309-310: Give the size distribution (e.g. mean diameter and standard deviation) generated and define what you consider respirable range. Explain why the respirable range is of interest here.

We propose to change the text as follow:

"The accelerated air jet breaks up the liquid into droplets. The aerosol generated by this process is sprayed downwards inside the jar where the larger droplets are collected on the liquid surface due to impaction as they cannot make the U-turn while the finest droplets are forced up through the outlet tube on top of the BLAM lid. The result is a very fine mist, well within the respirable range (i.e. with diameter smaller than 10 ïA■m) and with narrow size distribution. The size distribution, immediately after the injection of physiological solution (with or without bacteria) in ChAMBRe, shows a mean value of 0.45 ïA■m with a standard deviation of 0.25ïA■m".

This information, however, is just a typical figure since the actual size depend on the solution we nebulize according to the type and concentration of the solute. The interest in the respirable range is triggered by health issues. We'd consider this point quite clear but we can easily add a sentence in the text may be in the introduction.

Page 9 line 321-329: How do you avoid contamination of the Petri dishes, as they seem to be exposed to laboratory air?

See page 10 point e) The sterilizing UV lamp (ozone free, see section 2.2) is switched on for 15 minutes before injection to guarantee the Petri dishes sterilization.

Page 9 line 335: Explain the gravitational settling method. What is the settling time distribution for the bacteria you studied? Compare the average settling times with the typical atmospheric residence times of those bacteria.

It is assumed that the living microorganisms present in the aerosol are deposited on the petri dishes by gravity without undergoing any stress, from those related to the permanence in the experimental setup atmospheric conditions. In this way, it can be assumed that the number of units forming colonies counted on a Petri dish is proportional to the number of aerosolized and suspended living microorganisms within the chamber and also to the concentration value of viable bacteria in the aerosol. Lee et al., 2002 suggest that the average aerodynamic diameters of generated E. coli and B. subtilis aerosols were 0.63 and 0.75 $\mu$m respectively. If compare these data with data

obtained with NaCl solution to determine particles life time in chamber, the bacteria life time is aspect to be around five hours. The mean global residence time calculated by Burrows et al., 2009b, lie between 2 and 15 days for bacteria traces.

This explanation will be included in the revised manuscript.

Page 9 line 336: Define in which respect you consider this method as efficient.

We understand the statement is too crude. Actually, during the pilot experiment described in Brotto et al., several methods to collect viable bacteria had been preliminary testes (including) filtration and impaction but the "gravimetric" collection on petri dishes was the best to keep bacteria alive and count the formed colonies. This was the background of the term "efficient". We propose to change the sentence in this way:

. . .proven to be a very suitable way to collect and count viable bacteria colonies (Brotto et al., 2015).

Page 9 line 338: Describe how the filter samples are collected.

The sampling was performed by exposing filters to the stream of aerosol coming out of the nebulizer, through a secondary flange connected at the chamber.

This sentence will be included in the revised manuscript.

Page 9 line 340: What do you mean with "tendency to aggregation"? Do you refer to sampling artifacts or to coagulation?

It is referred to bacteria tendency to form aggregates and biofilms in response to stress conditions.

We propose to change the sentence in the following way:

. . .ideal to study the morphology of cells and possible bacteria aggregates (e.g. biofilm formation) by scanning electron microscopy (Capannelli et al., 2011).

Page 9 line 348: Explain what kind of filter unit you used.

This is a very simple tool, quite common in chem. labs. (Fig. 3-4). Should we describe it in detail?

Page 10 line 369: Give an uncertainty for the estimated living bacteria concentration in the chamber e.g.: (10ËĘ5 ± ??) CFU mËĘ-3 = 0.1 CFU cmËĘ-3. Compare the number concentration of living and dead bacteria. Compare the number and size (mass) concentrations of the aerosol particles measured with the dead and living bacteria concentrations.

At the moment our equipment cannot discriminate between the number of live and dead bacteria, because our sampling method on solid petri dish allows us to evaluate only the cultivable fraction of aerosolized bacteria in the chamber. The 10ˆ5 values is given just to inform on the OoM of the concentration of injected bacteria. Actually, we refer and anchor our experiments to the CFU/ml in the BLAM solution which are known with the accuracy reported in Tables 2 and 4.

Page 10 line 373: What do you mean with "statistically compatible", significant?

We considered, in each petri dish, the number of counted colonies and its sqrt. value (i.e. the SD according to Poisson distribution) and we could observe that, within the interval delimited by the SD values, the counts in the four petri dishes were in agreement (i.e, statistically compatible) in each experiment.

We propose to change the sentence in:

turned out to be statistically compatible (i.e. within the interval delimited by the statistical uncertainty, the counts in the four petri dishes were in agreement).

Page 10 line 377: Do you mean no significant effect related to RH? Would you expect a significant effect related to the variations of relative humidity?

Yes, because for Escherichia coli a R.H at least 70% was essential for the success of the experiments (see section 5.2). We chose gram negative E. coli and gram positive B. subtilis because of the difference in their cell wall structure. Gram positive B. subtilis

has a single, relatively thick, and hardy cell wall, while gram negative E. coli has double layers and a soft cell wall (Madigan, Martinko, & Parker, 2000, Chapter 3. Brock biology of microorganisms (9th ed.)). From this difference in cell walls, we suppose a difference in the aerosolized behavior, and the results seems to confirm this prediction (FESEM micrographs).

How could this be related to the residence time or drying time of the bacteria?

Lee et al., 2002 suggest the changes of aerodynamic diameters of the aerosols as a function of the relative humidity. E. coli and B. subtilis aerosols grow significantly above 85% relative humidity and E. coli aerosols grow more than B. subtilis (Fig. 5-6). Our range of R.H. was always under this percentage, so we do not aspect a difference in the residence time between the experiments.

Page 10 line 382: Explain how the relative humidity in the chamber was increased and what you mean with the environmental value.

The relative humidity inside the chamber was controlled by changing the working condition of the humidifier. "registered environmental value" means the relative humidity value recorded that day in the laboratory surrounding the simulation chamber.

We propose to change the sentence in:

the environmental value recorded in the laboratory, by changing the working condition of the humidifier.

Page 11 line 395: Indicate if the uncertainty in the slope of the correlation ($\pm 5$ %?) includes the uncertainties of the individual measurement values in the plot. Replace "about 5 %" and "around 10 %" with well-defined values.

The slope uncertainty is determined including those of single points. We'll correct the text with values better defined (i.e.: 7% for B. subtilis and 4% for E. coli).

Page 11 line 399: Explain how the use of the optical density measurements influences

the uncertainty of the cell quantification.

We agree the sentence is not clear enough. Actually, our aim here was simply to underline that the results reported in the manuscript are encouraging in this direction too (that's why se said: seems therefore sufficiently adequate) but the possibility to firmly related OD to collected CFU remains to be verified. We propose to change the sentence as follows:

For E. coli suspension, the evaluation of the microbial concentration through the fast and simpler control of the optical density, seems possibly be accurate enough to perform controlled experiments, provided an adequate calibration of the whole procedure is carried out.

Page 11 line 404-408: Double check these observations by comparison with the particle measurements in the chamber.

Actually, this check cannot be performed at the moment since the particle sizers counts during the experiments are totally dominated by salt particles.

Page 11 line 410: The conclusions should be reformulated and extended including a comparison of typical atmospheric residence times for bacteria with those that can be achieved within ChAMBRe.

We propose to add in the conclusions the following statement:

Residence times of viable bacteria in ChAMBre are less than 5 hours, much shorter than the generic residence time in the open atmosphere. However, previous literature studies (Brotto et al., 2015) suggest that such time window is long enough to observe the effects (i.e. viability change) of bacteria exposure to air pollutants. The assessment of such effects is objective of the fore coming studies at ChAMBRe.

Table 1: Explain the range of relative humidities.

The range shows the minimum and the maximum value of relative humidity, measured

inside the chamber during the experiment.

Tabel 3: For which times during the experiments did you determine the relative humidities and temperatures?

Actually, these parameters are measured in continuous.

Tabel 4: Can you estimate the ratio of CFU vs. non-CFU deposited on the Petri dishes e.g. based on the bacteria concentrations, sedimentation rate and area of the Petri dishes?

No, we can't. Actually, we could say that the ratio could arrive to be around 10 but this figure would be based on too much speculative arguments and we prefer to postpone this kind of consideration to further experiments performed with other collection methods (presently in the set-up phase).

Figure 1: Explain the different parts of the chamber in the caption. Is there any air guiding tube surrounding the fan? Is the fan efficiently mixing the upper and typically warmer part of the chamber? What is the typical vertical temperature gradient? Would it be possible to heat the chamber to 37_C?

There is no guiding tube. The mixing time has been measured connecting the analyzer in the upper part of the Chamber (see above our answer to question on Page 6 line 202). We have at the moment just one T sensor and we cannot measure the T gradient however, considered that the stainless chamber is relatively small and it is inside a climatic room we do not expect large variations. It is certainly possible to heat the chamber and we are working to implement a stable and reliable system.

Figure 3: Indicate RH and temperature for the experiment in the caption.

These values will be added in the caption: 47%, 21 °C.

Figure 4: Include uncertainties for the individual data points. Compare the optical density measurement to the CFU data and/or literature data.

Uncertainties are at the 5% level, they will be indicated in the revised figure. The relationship between OD and CFU is quite standard (at least in micro-biology), an example for E. Coli is reported in Fig. 7 (red: theory, blue: measured).We'd not consider to add this kind of information (well known in literature) in the text but we could obviously add a Figure.

Figures 5&6: Indicate if the uncertainties for the individual data points are included in uncertainties of the slopes.

Please, see our answer to comment at Page 11 line 395

Technical corrections:

Technical corrections will be considered and added to the revised manuscript.

Page 1 line 16: . . .processes at realistic but controlled conditions. Done. Page 1 line 21: . . .is made of stainless steel. . . Done. Page 1 line 22: . . .10 to 2 hours. Done. Page 1 line 24: . . . have impact on several levels as: . . . Done. Page 1 line 25: . . ., and geochemistry. Done. Page 2 line 40: . . .and maintain their pathogenic potential, . . . Done. Page 2 line 41-42: check wording Page 2 line 44: . . .chemical, and biological properties. . . Done. Page 2 line 61-63: Reformulate sentence without brackets. Done. Page 3 line 109: . . .height. Done. Page 4 line 114: . . .designed to move specific samples inside the chamber as described. . . Done. Page 4 line 120: . . .failure it automatically closes in less than one ms, . . . Done. Page 4 line 127: . . .with four metallic arms of 25 cm length each. . . Done. Page 4 line 131: . . .and an accuracy of $\pm$10% of its reading. . . Done. Page 4 line 133: The pressure transducer contains. . . Done. Page 5 line 185: . . .concentrations are monitored. . . Done. Page 7 line 236-238: Reformulate the sentence in correct English.

The sentence will be modified as follows:

In this section we describe the standard methodology developed for the bio-aerosol experiments (injection, collection and storage) and the related experimental conditions,

[Figure]

that should be representative of the typical environmental ones.

Page 7 line 255: Reformulate the sentence in correct English.

The sentence will be modified as follows:

The same culture preparation technique was applied at both the bacterial strains, in order to minimize experimental variations.

Page 7 line 257: . . .prior to the injection. Done. Page 7 line 268: In both cases,. . . Done. Page 8 line 278: The cultivable cell concentration. . . Done. Page 8 line 288: . . .was prepared for nebulization. . . Done. Page 8 line 290: . . .was diluted (. . . Done. Page 8 line 306: . . .with a cavity depth and a cone diameter of . . . Done. Page 8 line 313-314: . . .completely separate the cylinder, which can be connected to the main chamber or. . .. Done. Page 9 line 316: This volume can be evacuated through a by-pass to the. . . Done. Page 9 line 325: Valve V2 is closed and the volume inside the pipe is flushed with clean air from the chamber. Done. Page 9 line 332: . . .bacteria have been injected. . . Done. Page 9 line 33: After exposure to the chamber atmosphere, . . . Done. Page 9 line 345: . . ..were not done in this case as the study. . . Done Page 10 line 371: . . ..contributions. Done. Page 10 line 374: . . .appears to be adequate. . . Done. Page 11 line 402: . . .second set of experiments providing the Gram-negative microorganisms a more. . . Done.
* * *
**Experiment to determine physical wall loss of particles**

The physical wall loss of particles in close vessels such as chambers is a key parameters that vary with the size of the considered particles. The wall loss rate as a function of the size is depending on 1/ the chamber shape, 2/ the mixing regime (especially for small particles), 3/ the density of the considered particles, 4/ the electrostatic state of the wall. It has been very well studied (McMurry and Grosjean, 1985; McMurry and Rader, 1985) and efficient parameterization are available (Crump et al., 1983; Crump and Seinfeld, 1981; K. Lai and Nazaroff, 2000)

A rather simple experiments provide straightforward procedure. The principle of these characterization experiments is to generate a polydisperse aerosol sufficiently diluted to neglect coagulation and sufficient inert to neglect condensation or evaporation. This can be done by the nebulisation of a saline solution (eg. NaCl or ammonium sulfate) or by the chemical conversion of gases (ozonolysis of small quantity of pinene, photolysis of SO2/ozone/water…) for sub-micronic particles or by mechanic generation for super-micronic particles. The total number concentration must be below 10^4 #/cc to minimize the collision probability and so the coagulation process. When this is achieved (and when the aerosol formation is negligible for chemical generation), one just have to follow the number size-distribution as a function of time (with a SMPS or a APS for sub-micronic particles, with Grimm$^{TM}$, Welas$^®$ or other OPC for super-micronic particles). Then one have to fit for each size-bins the decay with a first order law. This hypothesis is generally working well as bouncing or re-emission from the wall are often not too significant.

Being a basic knowledge about a chamber, it is recommend that such an experiment (for sub- and/or super-micronic depending on the use) should be done at least once for rigid non-electrostatic chamber and regularly for Teflon chamber

Crump, J.G., Flagan, R.C. and Seinfeld, J.H., 1983. Particle Wall Loss Rates in Vessels. Aerosol Science and Technology, 2(3): 303 - 309.
Crump, J.G. and Seinfeld, J.H., 1981. Turbulent deposition and gravitational sedimentation of an aerosol in a vessel of arbitrary shape. Journal of Aerosol Science, 12(5): 405-415.
K. Lai, A.C. and Nazaroff, W.W., 2000. Modelling indoor particle deposition from turbulent flow onto smooth surfaces. Journal of Aerosol Science, 31(4): 463-476.
McMurry, P.H. and Grosjean, D., 1985. Gas and aerosol wall losses in Teflon film smog chambers. Environmental Science & Technology, 19(12): 1176-1182.
McMurry, P.H. and Rader, D.J., 1985. Aerosol Wall Losses in Electrically Charged Chambers. Aerosol Science and Technology, 4(3): 249-268.

**Fig. 1.**

[Figure]

Fig. 2.

[Figure]

**Fig. 3.**

**Fig. 4.**

[Figure]

B.U. Lee et al. / Aerosol Science 33 (2002) 1721–1723

Fig. 1. The changes of aerodynamic diameters of aerosols as a function of the relative humidity.

**Fig. 5.**

[Figure]

Fig. 2. Hygroscopic growth factors as a function of the relative humidity.

**Fig. 6.**

[Figure]

**Fig. 7.**